# Mammary gland multi-omics data reveals new genetic insights into milk production traits in dairy cattle

Wentao Cai[1,2,3], John B. Cole[4,5,6], Michael E. Goddard[7,8], Junya Li[1], Shengli Zhang[3], Jiuzhou Song[2]*

1 Institute of Animal Science, Chinese Academy of Agricultural Sciences, Beijing, China, 2 Department of Animal and Avian Science, University of Maryland, College Park, Maryland, United States of America, 3 College of Animal Science and Technology, China Agricultural University, Beijing, China, 4 Animal Genomics and Improvement Laboratory, USDA, Beltsville, Maryland, United States of America, 5 Department of Animal Sciences, University of Florida, Gainesville, Florida, United States of America, 6 Department of Animal Science, North Carolina State University, Raleigh, North Carolina, United States of America, 7 Agriculture Victoria, AgriBio, Centre for AgriBiosciences, Bundoora, Victoria, Australia, 8 Faculty of Veterinary & Agricultural Science, The University of Melbourne, Parkville, Victoria, Australia

* songj88@umd.edu

## Abstract

Although many sequence variants have been discovered in cattle, deciphering the relationship between genome and phenome remains a significant challenge. In this study, we identified functional classes, including mammary-specific genes, lactation-associated genes, novel long non-coding RNAs, miRNAs, RNA editing sites, DNA methylation, histone modifications, and expression quantitative trait loci. We estimated their contributions to genetic variance for milk production traits using 3 million variants in 23,566 Holstein bulls. Sequence variants in the 5'-UTR, synonymous, and splicing regions disproportionately contributed to genetic variance of milk production traits compared to other genomic regions. Genes specifically expressed in the mammary gland, particularly those active in lactating tissue (e.g., *GLYCAM1*, *DGAT1*), account for significantly more genetic variance of milk production traits than specific genes from non-mammary tissues. We identified 8,560 differentially expressed genes (DEGs) between lactating and non-lactating tissues. Among these, both up-regulated and small-fold changes of down-regulated DEGs exhibited greater genetic variance enrichment of milk production traits than other genes. Mammary enhancers (e.g., H3K27ac, H3K4Me1) explained more variance than repressive elements, while small changes in DNA methylation level (≤0.2) contributed more variance than that with larger changes (> 0.2). Notably, lactation-associated RNA editing sites in mammary explained more variance for milk production traits than expected by chance. We proposed a novel miRNA prioritization strategy for selecting candidate miRNAs related to milk production traits, based on the overlaps between significant enrichment tests of miRNA target correlations and the relatively large variance explained by these

**Data availability statement:** o The 6,642 RNA-seq data supporting this study's findings is available from the NCBI SRA and CNCB database with accession numbers in S9 Table. The 12 small RNA sequencing data is available from the NCBI SRA with accession number PRJNA689373. The 86 histone modification data and six DNA methylation data can be accessed by SRA number PRJEB41939 and GEO number GSE106538, respectively. The genotype and phenotype data are owned by third parties and managed by the Council on Dairy Cattle Breeding (CDCB). To obtain access for research purposes, requests can be sent to João Dürr, CDCB Chief Executive Officer (joao.durr@cdcb.us). Alternatively, you can contact the CDCB at 301-712-9339 or visit their website at https://uscdcb.com/contact/ for data use. The data and analysis codes used to generate the figures and the mammary-associated functional classes have been uploaded and published on FigShare (https://doi.org/10.6084/m9.figshare.27991175.v2). All other data are in the manuscript and its supporting information files.

**Funding:** o This work was supported by grants from the USDA/NIFA (MD-187584 to J.S.), Young Scientists Fund of the National Natural Science Foundation of China (32202652 to W.C.), the Youth Innovation Program of the Chinese Academy of Agricultural Sciences (Y2024QC09 to W.C.), China Agriculture Research System of MOF and MARA (CARS-37 to J.L.). The funders had no role in study design, data collection and analysis, decision to publish, or preparation of the manuscript.

**Competing interests:** The authors have declared that no competing interests exist.

targets. Additionally, we integrated these nine functional classes into the variance component analysis simultaneously, revealing that sQTLs, histone modification and DEGs showed the highest per-SNP variance enrichment. Finally, we constructed a new 624K SNP panel, which improved the reliabilities of genomic predictions by 0.22%. Dividing routine SNPs into two groups based on functional classes improved the reliabilities by 0.21%, particularly for milk protein percentage (0.68% improvement). Overall, incorporating prior biological knowledge of the mammary gland directly enhances our understanding of milk production's genetic architecture and improves the reliability of genomic predictions for milk production traits. This integrative approach establishes a paradigm for translating biological knowledge into agricultural genomics applications.

## Author summary

Milk production traits in dairy cattle are influenced by a complex interplay of genetic factors. While numerous sequence variants have been identified in cattle, linking these variants to specific phenotypes remains a significant challenge. Here, we incorporated prior biological knowledge, focusing on functional classes such as mammary-specific genes, lactation-associated genes, non-coding RNAs, miRNAs, RNA editing sites, DNA methylation, histone modifications, and expression quantitative trait loci. Analyzing 3 million variants in 23,566 Holstein bulls, we identified key variants and functional classes that contribute significantly to genetic variation in milk production. Notably, variants within the 5'UTR, synonymous regions, and splicing sites captured more genetic variance than other genomic regions. Additionally, our results highlighted the importance of lactation-up-regulated and down-regulated genes and lncRNAs in explaining genetic variance. We also proposed a novel strategy for candidate miRNA selection. Our findings demonstrate that integrating prior biological knowledge into genomic prediction models can significantly improve their accuracy, providing deeper insights into the genetic architecture underlying milk production in dairy cattle.

## Introduction

High-throughput technologies have revolutionized animal genetics research and enabled the creation of multi-omics data, encompassing genomics, transcriptomics, proteomics, epigenomics, and metabolomics. Multi-omics data can facilitate the collection of molecular phenotypes, thereby accelerating the deciphering of genetic mechanisms underlying complex milk production traits [1,2]. Understanding genome-to-phenome using functional molecules is crucial for molecular precision dairy breeding since these molecules are involved in key biological processes and affect milk protein and fat synthesis [3,4]. Omics technologies have been widely

used to identify lactation-related protein-coding genes [5,6], miRNAs [7,8], long non-coding RNAs (lncRNA) [9], DNA methylation regions [10], and histone modifications [11], which may provide greater insight into biochemical and genetic mechanisms of milk synthesis in the mammary gland.

Whole-genome sequencing overcomes some drawbacks of SNP genotyping arrays and may enhance the effectiveness of variant detection, genome-wide association studies (GWAS), and genomic prediction. The 1000 Bull Genomes Project offers a large database that can be leveraged for imputing genetic variants in cattle, serving as a valuable resource for genomic prediction and GWAS [12]. Various strategies have been employed to replace the routine SNP panel with imputed variants, which has led to marginal improvements in predictive accuracy [13,14]. One effective approach involves narrowing down the vast array of variants to a more manageable subset by focusing on functional classes known to be associated with complex traits. This approach increases the likelihood that the selected variants will significantly influence these traits. Additionally, annotating genetic variations into functional classes has proven beneficial for establishing associations between these functional classes and complex traits in previous cattle studies [15–17]. In case of dairy traits, the variants at the splice sites explained the highest proportions of phenotypic variance for milk production traits per variant. Conversely, lncRNA, miRNA target sites, and transcription factor binding sites (TFBS) captured modest to large proportions of the variance [17]. Based on the predicted heritability of each variant across functional and evolutionary categories, such as genomic location, and selection signatures, Functional-And-Evolutionary Trait Heritability (FAETH) scores were proposed to provide effective biological priors for GWAS and genomic prediction [15]. However, these functional classes were defined from annotation of many different cattle tissues. The genome partitioning of genetic variation by functional classes specific to mammary tissue has not been done in previous studies. Furthermore, novel functional classes, such as RNA editing associated with milk production traits, have yet to be thoroughly investigated in cattle. It has been suggested to prioritize likely causal markers as prior information in genomic prediction models to enhance prediction accuracy [18,19]. This can be achieved by assigning higher weights to individual or groups of markers within the genomic prediction model. These approaches facilitate the use of biological priors, such as BayesRC [20], MultiBLUP [21], and GFBLUP [18]. By employing differentially expressed genes as priors, the accuracy of genomic prediction for mastitis and milk production traits was enhanced by 3.2% to 3.9% using GFBLUP compared to GBLUP [19]. The application of BayesRC, as compared to the standard BayesR approach, resulted in improved accuracy of genomic predictions for milk production traits. This improvement was particularly noticeable when there was a greater genetic distance between the training and validation populations [20]. However, the magnitude of improvements to prediction accuracy was not as large as expected, mainly because the biological information may come from irrelevant tissues. As we know, biological information from different organs cannot contribute equally to specific production traits of interest. As an important functional organ in dairy cattle, mammary gland contains exactly functional information of milk synthesis, secretion, and production, which can be used to improve the accuracy of genomic prediction.

In this work, we hypothesized that multi-omics information focused on the mammary gland can explain a larger proportion of genetic variation than expected by chance from any biological data, which will improve the accuracy of genomic prediction and provide mechanistic clues about the milk synthesis process. This study of the mammary gland aims to 1) identify and define the sequencing segments and candidate biological molecules, such as coding/non-coding genes, RNA editing, DNA methylation, and histone modification, using mammary multi-omics data; 2) determine which functional indicators are most effective in predicting sequence variants with the highest probability of influencing milk production traits in dairy cattle; and 3) assess the increase in accuracy of genomic predictions that can be achieved using the prior biological information, compared with that from the non-prior information strategy in the same data set (Fig 1).

## Results

### Genome partitioning of genetic variance

For genome partitioning analysis, we used 23,566 Holstein bulls that possess highly reliable breeding values, specifically deregressed predicted transmitting abilities (PTAs), across seven traits related to production and reproduction (S1 Table).

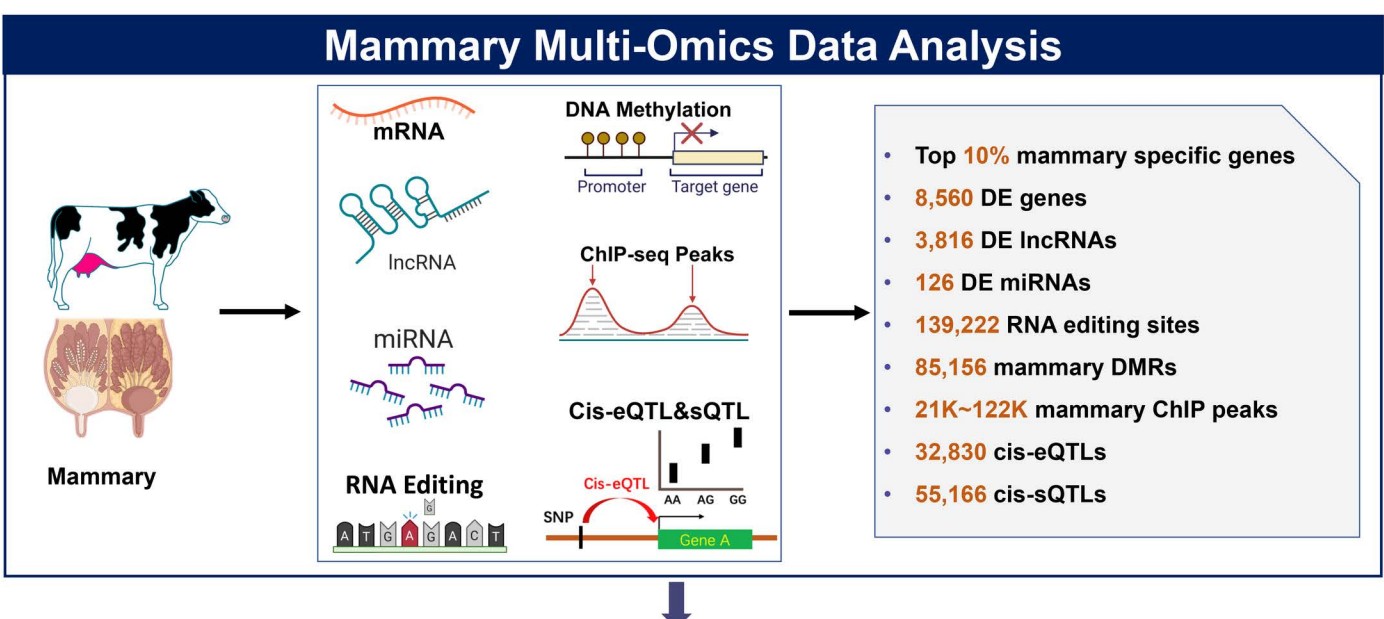

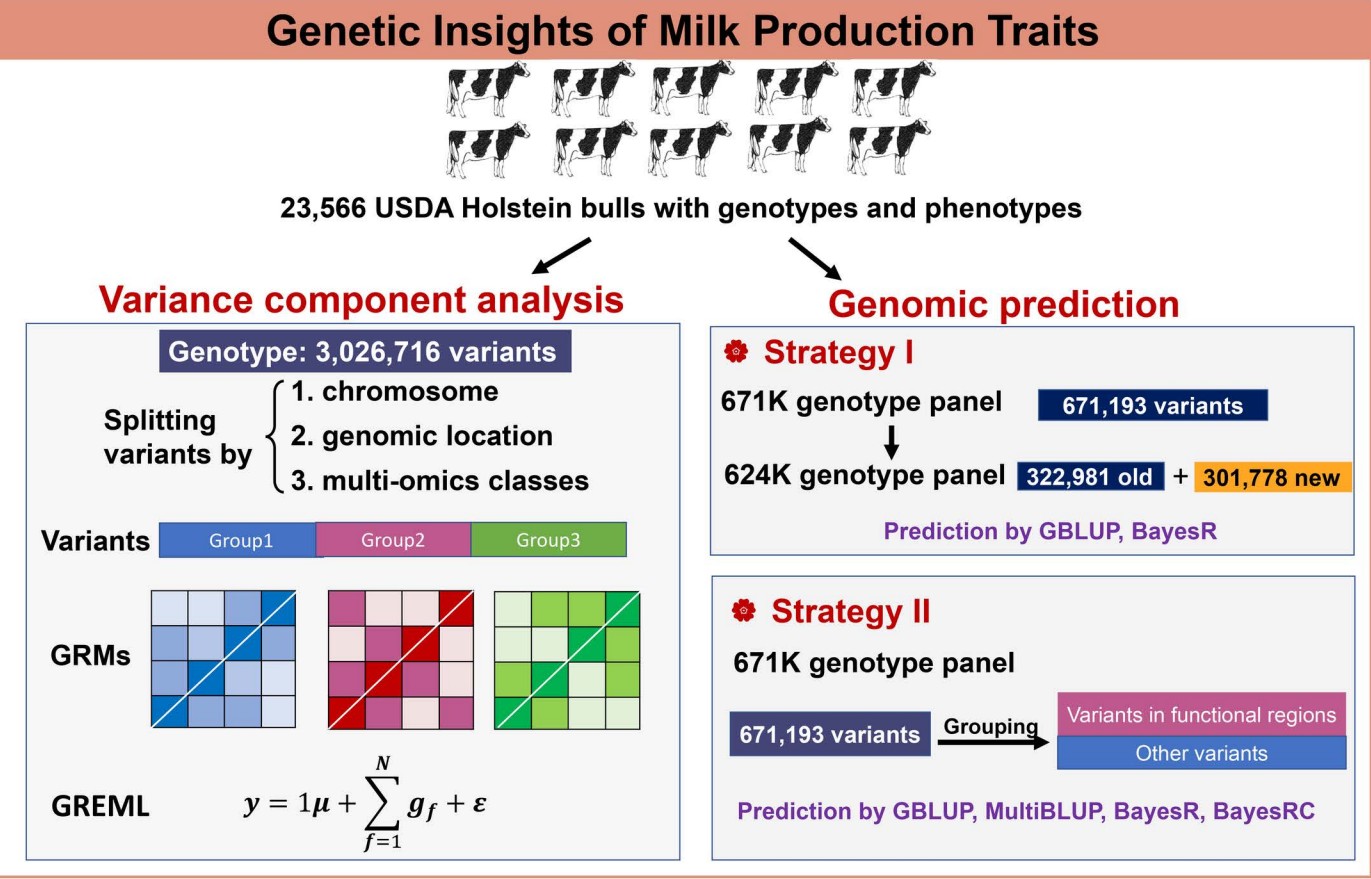

**Fig 1. Overview of the analysis.** The functional classes, including mammary-specific genes, lactation-associated genes, novel long non-coding RNAs, miRNAs, RNA editing sites, DNA methylation, histone modifications, expression quantitative trait loci (QTL), and splicing QTL, were defined using mammary multi-omics data. Variance component analysis of milk production traits was performed based on variants split using GREML in 23,566 Holstein bulls. Two strategies for genotype modification were applied to improve the reliability of genomic predictions.

We computed the genetic relationship matrix (GRM) and incorporated it into a mixed linear model (MLM) to quantify the proportion of variance attributed to autosomal SNPs for traits. We estimated that 75.1%, 76.1%, 71.1%, 89.2%, and 87.7% of phenotypic variation for milk yield (MY), protein yield (MPY), fat yield (MFY), protein percentage (MPP), and fat percentage (MFP), respectively, were tagged by autosomal SNPs. Other traits were shown in S1 Table. To allocate the total genetic variance among specific chromosomes, we constructed separate GRM using the SNPs from each autosome. These individual GRMs were then incorporated simultaneously into a joint model to effectively partition the genetic variance across the chromosomes. We detected a strong linear correlation between the proportion of variance explained by each chromosome and the length of the respective chromosome for both MY and MPY (Fig 2A and 2C). Chromosome 14 can explain a relatively large proportion of variance for milk production traits (Fig 2A-E), particularly with 23.9% of the variance attributed to MFP.The proportion of variance of MPP was largely captured by chromosome 6 (Fig 2E). The proportion of variance explanation of reproductive traits is less linearly correlated with chromosome length (S1 Fig).

To quantify genetic variation explained by genomic regions, we partitioned the variance explained by all the SNPs onto intergenic, synonymous, missense, intron, 3'-UTR, 5'-UTR, downstream, upstream, and splicing regions of the whole genome. Due to intergenic and intronic regions covering 57.8% and 27.9% of the total SNPs, respectively, their variance proportions were relatively large (S2 Fig). After adjusting the SNP numbers of each region, we observed that variants

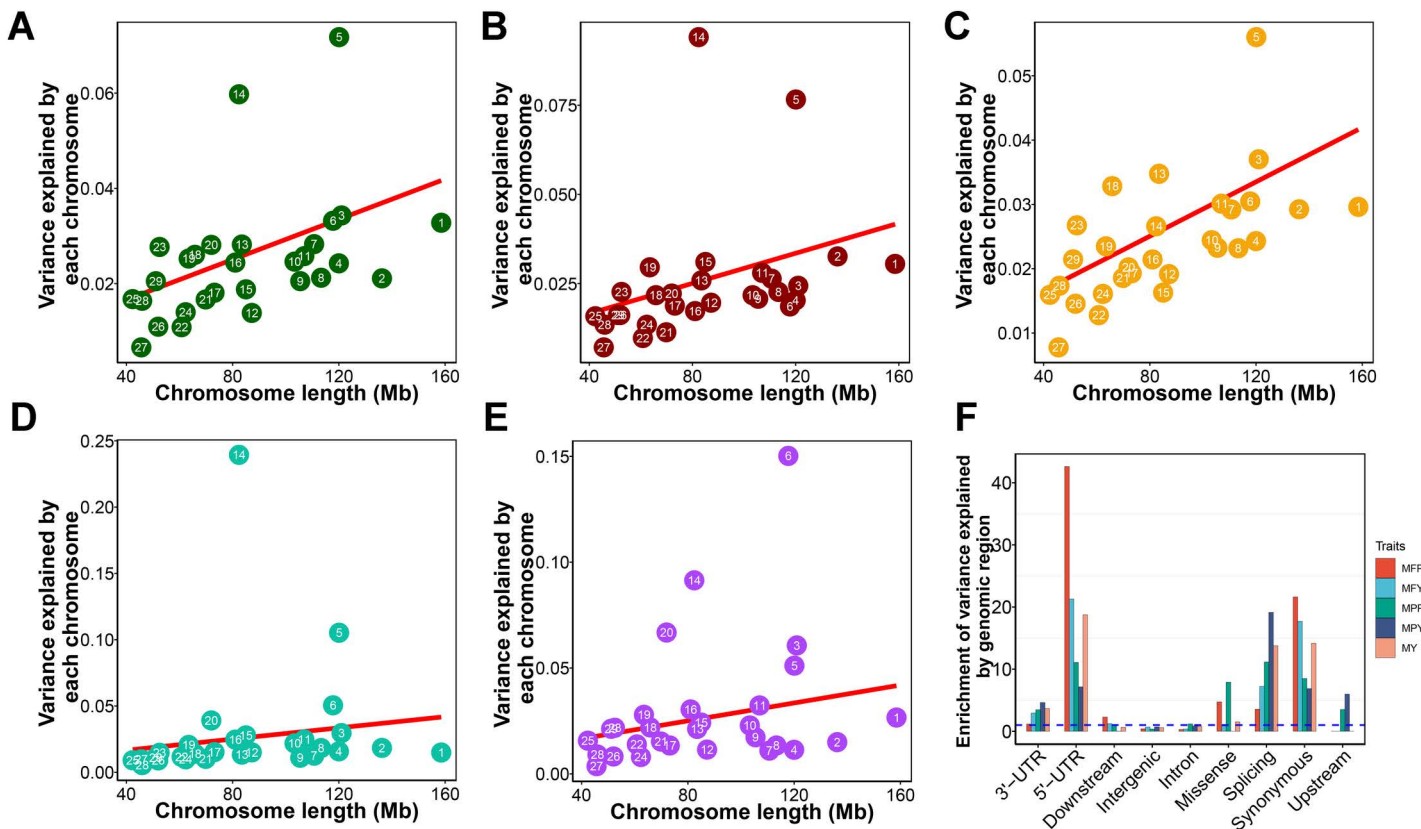

**Fig 2. The proportion of variance explained by chromosomes and genomic regions.** The proportion of variance explained by each chromosome against chromosome length is shown for (A) milk yield (MY), (B) milk fat yield (MFY), (C) milk protein yield (MPY), (D) milk fat percentage (MFP) and (E) milk protein percentage (MPP) by joint analysis The numbers in the circles and squares are the chromosome numbers. The regression adjusted R² (P-value) were 0.167 (0.017) for MY, 0.074 (0.083) for MFY, 0.335 (5.9 × 10⁻⁴) for MPY, -0.016 (0.465) for MFP, and 0.047 (0.135) for MPP, respectively. (F) The enrichment of the variance explanation of each class of genomic region in five traits. The enrichments were calculated using the odd ratios between the proportion of variance explained and the proportion of SNP number.

located in the 5'-UTR, synonymous, and splicing regions accounted for more genetic variance than those found in inter-genic regions, introns, and downstream regions (Fig 2F). Upstream variants could explain more genetic variance for MPY and MPP, while the missense could explain more genetic variance for MFP and MPP.

**Mammary specific genes**

To identify mammary-specific genes, we first obtained a *t*-statistic for each gene across 6,642 RNA-seq data sets to measure its expression specificity in the mammary tissue. A high *t*-statistic means the gene is more specific to a given tissue. We observed the top specific genes in lactating mammary gland (*GLYCAM1*), non-lactating mammary gland (*ENSBTAG00000049824*), mammary fat pad (*RPL23A*), mammary parenchyma (*ENSBTAG00000012491*), and milk cell (*GLYCAM1*). We observed highly positive correlations in the t-statistics of genes among non-lactating mammary gland, mammary parenchyma, and mammary fat pad. In contrast, lactation mammary gland had a moderate correlation with milk cells based on their gene *t*-statistics (S3 Fig). We defined tissue-specific genes for each tested tissue based on the rank of t-statistics (top 10%). Additionally, we partitioned the genetic variance in milk production explained by SNPs into tissue-specific genes or their flanking 5 kb regions (Fig 3A). As expected, the lactating mammary gland-specific genes explained more genetic variance of milk production traits (MFP and MPP) than the other four mammary tissues (Table 1). Otherwise, in non-mammary tissue, we found the liver captured consider-able genetic variance in milk production traits, especially for MFY and MFP.

**Lactation associated genes**

We detected 20,379 expressed genes (FPKM > 0.1 in at least two samples) in 103 biopsy mammary glands. These samples could be separated into different lactation stages based on expression levels using PCA (Fig 3B). We identi-fied 8,560 significantly dysregulated genes between the non-lactating period and lactation using DESeq2 with adjusted *P*-value < 0.05, including 3,790 up-regulated genes and 4,770 down-regulated genes in lactation (S2 Table). The list of the up-regulated genes contained milk protein and fat genes such as *CSN1S1, CSN1S2, CSN2, CSN3, LGB, LALBA, DGAT1, FASN, SCD, ABCG2,* and *SREBF1.* Functional annotation revealed that the up-regulated genes were involved in milk metabolism and energy metabolism, such as metabolic pathways, oxidative phosphorylation, protein processing in the endoplasmic reticulum, protein export, and the AMPK signaling pathway. The top lists of down-regulated genes were *BOLA-DQB, LAMA1, LGR6, TMEM213,* and *NF-M.* The down-regulated genes were associated with disease and immune-related function, such as HTLV-I infection, pathways in cancer, cell cycle, cell adhesion molecules (CAMs), and cytokine-cytokine receptor interaction (S3 Table).

To assess the contribution of genetic variance by differentially expressed genes (DEGs), we classified variants into two groups based on whether they were located within DEGs or their flanking 5 kb regions, or not. We applied a simi-lar classification strategy for non-DEGs. Our analysis revealed that DEGs could explain large proportion of variance for milk production traits (Fig 3C and Table 1) for MFP (0.72) and MPP (0.6), moderate proportion of variance for MY (0.39), MFY (0.43), and MPY (0.26). After correcting the variant number in each group, we observed that proportion of variance explained per variant was larger for the DEGs than non-DEGs in all traits (Fig 3D). To better understand the genetic variance explained by DEGs, we divided these DEGs into eight groups based on the fold change values (i.e., ≤−8, −8~−4, −4~−2, −2~−1, 1~2, 2~4, 4~8, and ≥8). In addition, we built two other groups based on variants located in genes that were not differential expressed (non-DEGs) or located outside of any genes (others). In total, 10 genomic groups were defined. We detected that the DEGs with small foldchange (|fold change| < 2) captured more genetic variance than other DEGs genes (S4 Fig). This phenomenon was primarily attributed to the *DGAT1* and its neighboring genes within the small-fold change groups. To eliminate the effects of *DGAT1*, we put the proximal SNPs within the *DGAT1* (including its flanking 1 Mb regions) as covariates. Then we detected genetic variance explained per variant was very high for these up-regulated DEGs in lactation, especially for fold change >8 (Figs 3E and S5). For down-regulated genes, the genetic variance explained per variant was only enriched in the group with foldchange −2~−1. These findings offer support for

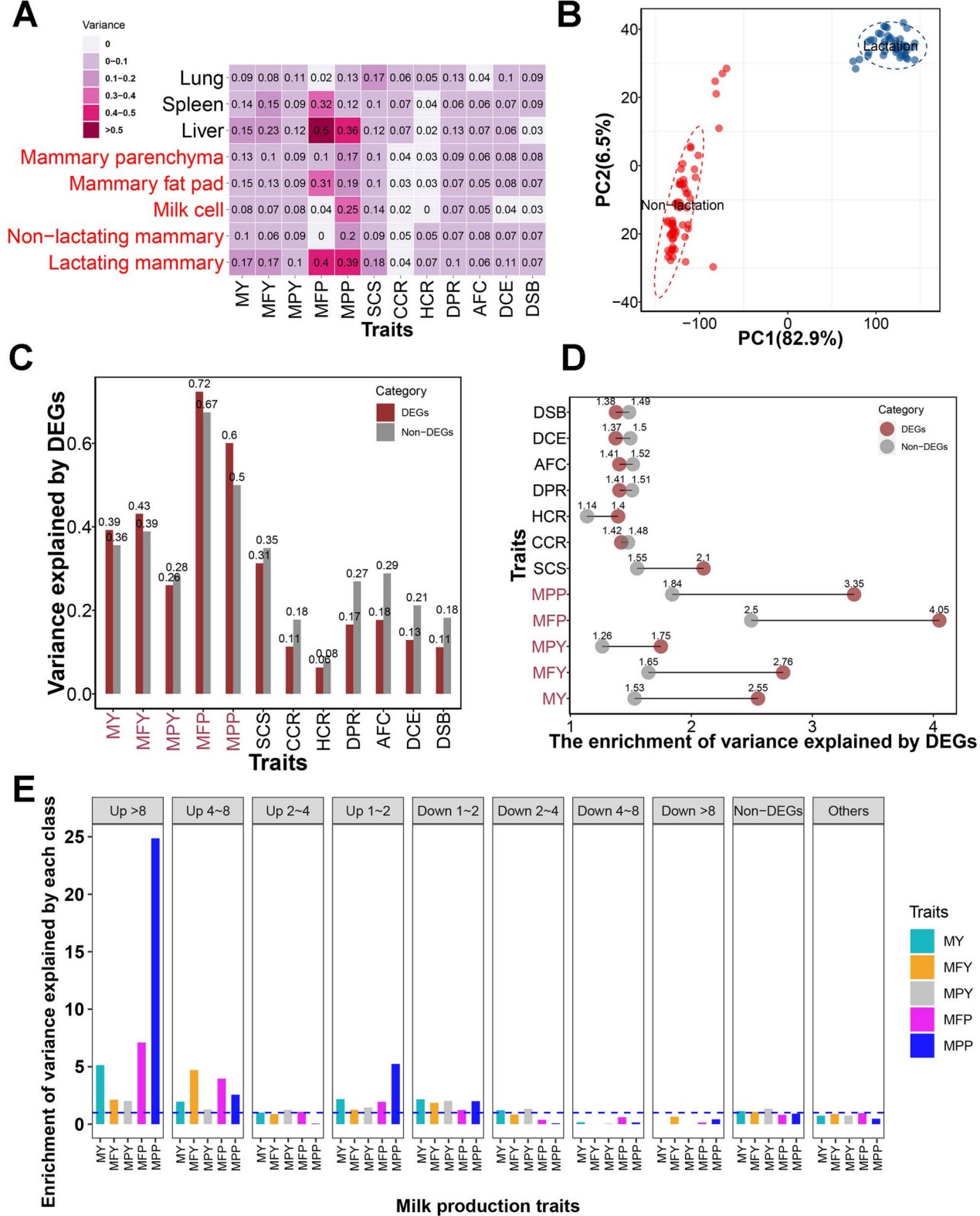

**Fig 3. The proportion of variance explained by specific genes and lactation associated genes.** (A) The proportion of variance explained by tissue-specific genes for milk yield (MY), milk fat yield (MFY), milk protein yield (MPY), milk fat percentage (MFP), milk protein percentage (MPP),

somatic cell score (SCS), cow conception rate (CCR), age at first calving (AFC), heifer conception rate (HCR), daughter calving ease (DCE), daughter still birth (DSB), and daughter pregnancy rate (DPR). (B) Sample clustering using PCA based on gene expression levels. (C) The proportion of variance explained by differentially expressed genes (DEGs) between lactation and non-lactaing period. (D) The enrichment of genetic variance explained by DEGs between lactation and non-lactaing period. The enrichments were calculated using the odd ratios between the proportion of variance explanation and the proportion of SNP number. (E) The enrichment of genetic variance is explained by different fold-change groups for milk production traits after correcting *DGAT1* regions.

**Table 1. The proportion of variance explained for each functional class across five milk production traits using two genomic relationship matrices (GRMs) model, which fitting each functional class (e.g., DEGs, miRNAs) separately against the non-functional SNP background and their respective GRMs.**

| Functional class | Variants number | MY | MFY | MPY | MFP | MPP |
|---|---|---|---|---|---|---|
| Specific genes ± 5Kb | 248,903 | 0.078 | 0.098 | 0.047 | 0.307 | 0.182 |
| DEGs ± 5Kb | 631,689 | 0.392 | 0.431 | 0.260 | 0.724 | 0.601 |
| DE lncRNAs ± 5Kb | 177,123 | 0.122 | 0.136 | 0.073 | 0.324 | 0.147 |
| DE miRNAs ± 5Kb | 1,796 | 0.026 | 0.033 | 0.003 | 0.142 | 0.065 |
| RNA editing ± 100Kb | 77,442 | 0.144 | 0.165 | 0.059 | 0.357 | 0.233 |
| DMRs | 61,661 | 0.177 | 0.237 | 0.109 | 0.418 | 0.197 |
| Enhancers | 183,768 | 0.337 | 0.385 | 0.243 | 0.591 | 0.403 |
| eQTLs | 32,830 | 0.128 | 0.185 | 0.073 | 0.322 | 0.147 |
| sQTLs | 55,166 | 0.203 | 0.231 | 0.124 | 0.439 | 0.270 |

Specific genes represent the top 10% of genes based on t-statistics of lactating mammary across 6,642 samples with 13 tissue categories. Differentially expressed genes (DEGs), Differentially expressed (DE) lncRNAs, and DE mRNAs represent differentially expressed genes, novel lncRNA, and miRNAs between lactating mammary and non-lactating mammary, respectively. RNA editing means the levels of RNA editing sites that were different or specif-ic between lactating and non-lactating mammary. DNA methylation regions (DMRs) represent differentially methylated regions between lactating and non-lactating mammary. Enhancers represent the histone modification (H3K27ac, H3K4Me1, and H3K4Me3) regions in lactating mammary. eQTLs and sQTLs were collected from the lactating mammary of cattle GTEx database [22]. Five milk production traits include milk yield (MY), milk fat yield (MFY), milk protein yield (MPY), milk fat percentage (MFP) and milk protein percentage (MPP).

the idea that the significantly up-regulated DEGs harbor genetic variants that have the potential to influence trait variation, making them a top priority for further investigation. Using the same dataset, we also identified 11,749 novel lncRNA transcripts in 5,176 lncRNA loci in mammary glands. Differentially expressed (DE) analysis revealed 3,816 lncRNAs were differentially expressed between the non-lactating period and lactation, including 1,295 lncRNAs up-regulated and 2,521 lncRNAs down-regulated in lactation. To assess genetic variance contributed by lncRNA, we captured proximal SNPs of DE lncRNAs, including their flanking 5 Kb regions (Table 1). We detected the down-regulated lncRNAs could capture 0.05~0.31 genetic variance for milk production traits, while the genetic variance explained by the up-regulated lncRNAs was limited (S4 Table). The genetic variance explained by down-regulated gene were MFP > MPP > MFY > MY > MPY.

## Lactation associated miRNAs

We identified 126 miRNAs that were differentially expressed between non-lactating period and lactation, including 69 miRNAs that were shown to increase and 57 miRNAs were observed to decrease in abundance at lactation. To identify potential candidate genes that could be regulated by specific miRNAs, we conducted an analysis to explore the negative correlation between the expression levels of a particular miRNA and the expression levels of all predicted mRNA targets of that miRNA in the corresponding samples. Our analysis indicated that 53 miRNAs exhibited a significant number of mRNA targets with negative correlations (Fisher's exact test P < 0.05, S5 Table). Of these, nearly two-thirds (35 miRNAs) of these 53 potentially functional miRNAs were also differentially expressed between the non-lactating period and lactation, indicating that these potentially functional miRNAs are more likely to be differentially expressed (Chi-square test, $P < 8.13 \times 10^{-16}$, Fig 4A). Interestingly, most of these DE miRNAs (32 of 35 miRNAs) were up-regulated in lactation.

To assess whether the variants in DE miRNAs accounted for a greater genetic variance, we captured proximal SNPs within the miRNA precursor regions, including their flanking 5 kb regions. We assumed all miRNA precursors were considered the same class and investigated the genetic variance explained by miRNA precursors for each trait (Table 1). We found variance explained per variant was larger for the DE miRNAs than all miRNAs for all traits except MPY (Fig 4B). These up-regulated miRNAs captured larger variance than down-regulated miRNAs (Fig 4C). These findings offer support for the notion that the significantly up-regulated miRNAs harbor genetic variants capable of impacting trait variation, making them a high-priority focus for further investigation.

We next checked whether the targets of trait-associated miRNAs are likely to explain more genetic variance for milk production traits. We grouped all targets of a specific miRNA together to form a miRNA targets class, and only SNPs located in targets (including 5 kb upstream/downstream) were included. We detected that the genetic variance of the milk production traits was not uniformly distributed along the genome but appeared to be enriched in a subset of target regions of lactation-related miRNAs. Variants in the miRNA-predicted target class for all five milk production traits captured more variance than expected, especially for the MFP trait, the average proportion of variance explained by DE miRNA targets was 0.17 (Fig 4D). We found the targets of lactation-associated miRNAs also can explain considerable variance for SCS, while the variance explained by miRNA targets for CCR was limited (Fig 4E). We observed the targets of these up-regulated miRNAs in lactation could capture a larger genetic variance than down-regulated miRNAs (S6 Fig). We identified candidate miRNAs for a trait based on the overlaps between significant enrichment tests of miRNA target correlations and the relatively large proportion of variance explained by these targets ($h_t^2 > 0.1$, S6 Table).

## Lactation-associated RNA editing

We identified 139,222 RNA editing sites in 12 mammary samples. Most RNA editing sites (98.9%) belonged to A-to-I (G) type. These A-to-I editing sites were located in 12,910 cluster regions. The average length of the RNA editing cluster regions was 92.1 bp and contained 10.7 editing sites. The largest cluster contained 295 RNA editing sites with a length of 1,538 bp. We found the majority of RNA editing sites were located in intergenic and introns further downstream or upstream. However, there were relatively few RNA editing sites detected in the 5'-UTR, 3'-UTR, and coding regions of genes (Fig 5A). Interestingly, the number of both RNA editing sites and events and the total RNA editing level were significantly lower in lactation compared to the non-lactating period (Fig 5B). We also observed the expression of *ADAR* was significantly down-regulated in lactation (adjusted P-value = $1.36 \times 10^{-6}$, Fig 5C). The *ADAR* had a strongly negative correlation with casein genes in expression (S7 Fig), which indicated that RNA editing in the mammary gland might be involved in milk protein synthesis related activities. To validate the identified RNA editing sites, we randomly selected eight regions and sequenced their DNA and complementary DNA (cDNA) from RNA using MAC-T cells by Sanger sequencing. All these eight regions were confirmed to contain RNA editing sites (S7 Table). For the regions in downstream of *TMED10*, we successfully validated 12 sites in the MAC-T cells (Fig 5I). Other validation results were shown in S8–S14 Figs.

A total of 41.4% of RNA editing sites were detected across multiple samples (S15 Fig). 5,569 and 1,902 RNA editing sites occurred in all six non-lactating and lactating mammary samples, respectively (Fig 5D). Of these, 668 (185 in lactation and 483 in the non-lactating period) were stage-specific, while 1,104 editing sites were common across stages. Functional enrichment analysis confirmed that the specifically edited genes in lactating mammary gland were involved in RNA phosphodiester bond hydrolysis, Golgi lumen, milk protein, response to dehydroepiandrosterone and 11-deoxycorticosterone, progesterone, and estradiol, as well as secreted and protein binding. In contrast, these specifically edited genes in non-lactating mammary gland were related to primary amine oxidase activity, amine metabolic process, aliphatic-amine oxidase activity, and phenylalanine metabolism (Fig 5D and S8 Table). The genes edited by stage-common editing sites were associated with complement activation, MHC II, immune response, and immunoglobulin receptor binding (S8 Table). These 12 mammary samples could be separated into two stages based on the editing levels of these 1,104 common sites (Fig 5E). We found that the patterns of editing levels for these common RNA editing sites were different between lactating and non-lactating

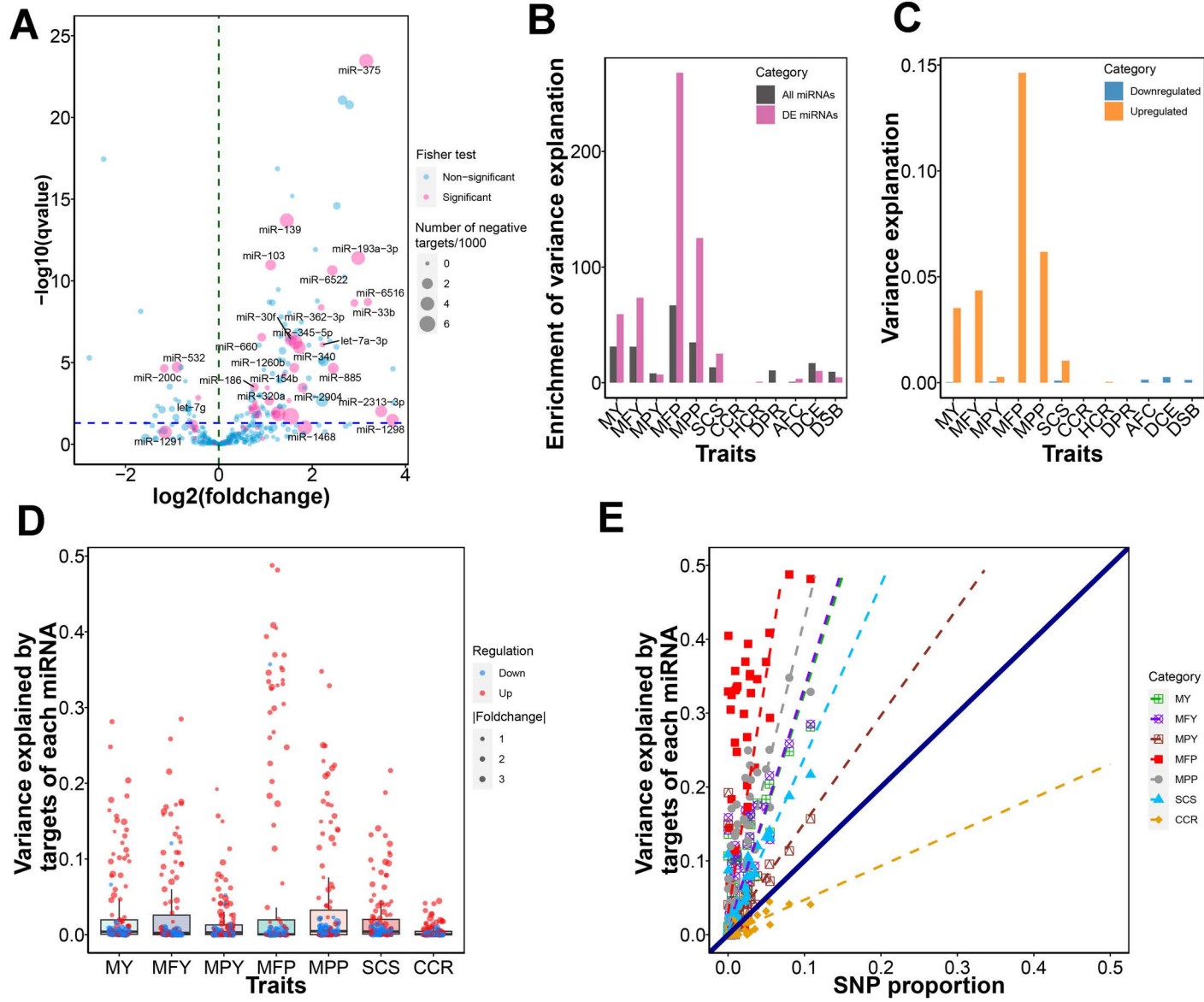

**Fig 4. The proportion of variance explained by lactation associated miRNAs.** (A) The volcano plot of mammary miRNAs. The x-axis represents the fold change of miRNA's expression between lactation and non-lactaing period with log2 transformed. The y-axis represents the adjusted P-value of each miRNA by differential expression analysis. Red color represents the miRNAs with significant P-value< 0.05 by fisher exact test based on whether an mRNA has a negative correlation with the intended miRNA or not versus whether it is a predicted target of the intended miRNA or not. The size of point represents the number of nagetively correlated targets of miRNA. (B) The enrichment of genetic variance is explained by differentially expressed (DE) miRNA precursors between lactation and non-lactaing period for milk yield (MY), milk fat yield (MFY), milk protein yield (MPY), milk fat percentage (MFP), milk protein percentage (MPP), somatic cell score (SCS), cow conception rate (CCR), age at first calving (AFC), heifer conception rate (HCR), daughter calving ease (DCE), daughter still birth (DSB), and daughter pregnancy rate (DPR). The enrichments were calculated using the odd ratios between the proportion of variance explanation and the proportion of SNP number. (C) The proportion of variance explained by up-regulated and down-regulated DE miRNA precursors. (D) The proportion of variance explained by the miRNA targets. The red and blue colors of point represent the down and up-regulated miRNAs. The size of point represents the absolute value of fold change. (E) The proportion of variance explained by the miRNA targets. The point represents each of the miRNAs. The x-axis represents the proportion of SNPs over the whole genome that are located in miRNA's targets; the y-axis represents the proportion of variance explained by each miRNA's targets.

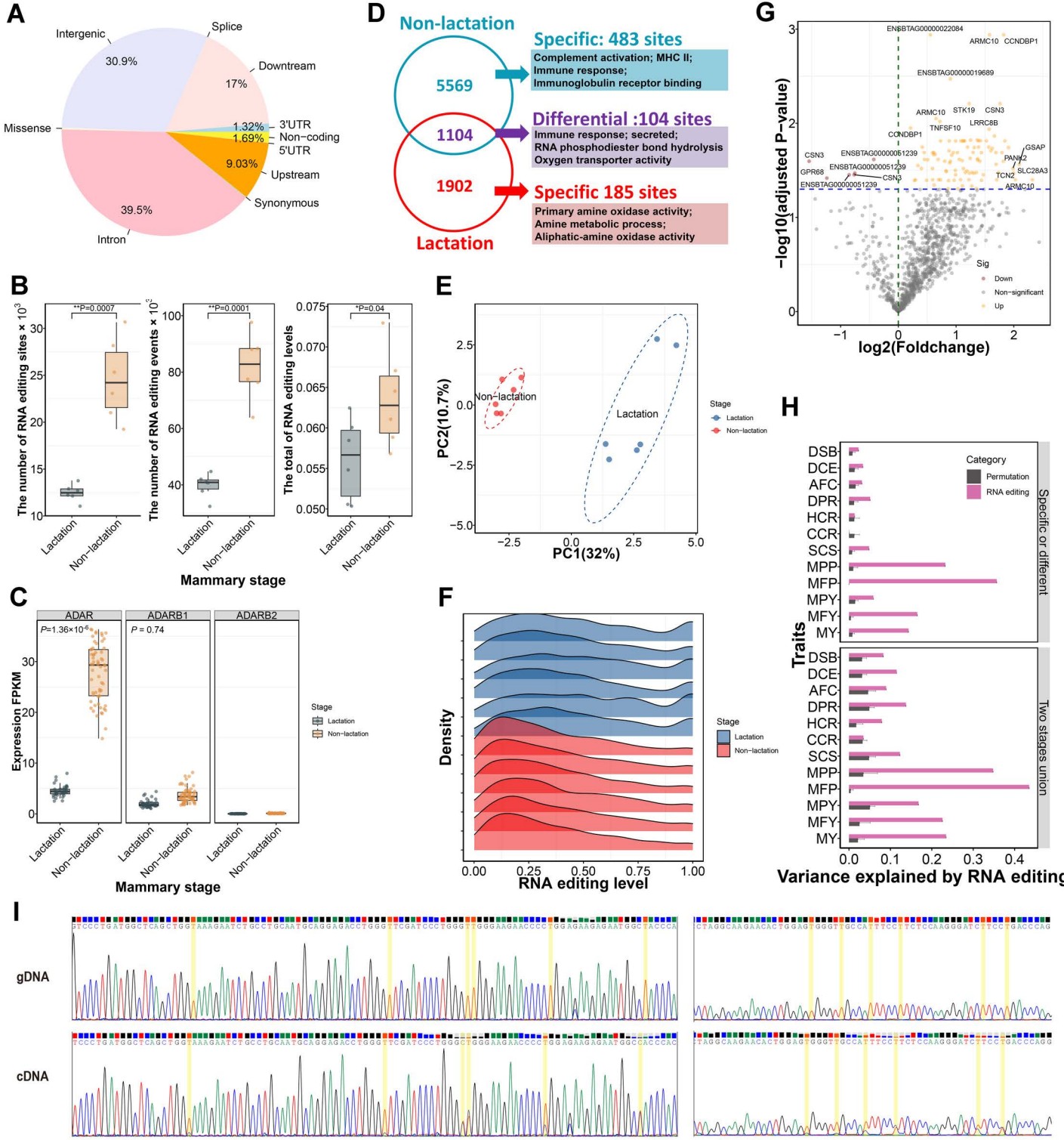

**Fig 5. The proportion of variance explained by lactation associated RNA editing sites.** (A) The distribution of RNA editing sites across different genomic regions. (B) The number of RNA editing sites and events, as well as RNA editing level compared between two mammary stages. (C) The expression of ADAR, ADARB1, and ADARB2 in two mammary stages. (D) The veen plot of RNA editing sites between lactation and non-lactation. The

functional annotations of lactation-specific, non-lactation-specific and differential RNA editing sites are shown. (E) Sample clustering using PCA based on the editing levels of these 1,104 common sites. (F) The patterns of editing levels for these common RNA editing in two mammary stages. (G) The volcano plot of these common RNA editing sites. The x-axis represents the fold change of RNA editing level for each editing sites between lactation and non-lactaing period with log2 transformed. The y-axis represents the adjusted P-value of each RNA editing site by differential expression analysis. (H) The proportion of variance explained by two stages union and stage-specific or different groups for milk yield (MY), milk fat yield (MFY), milk protein yield (MPY), milk fat percentage (MFP), milk protein percentage (MPP), somatic cell score (SCS), cow conception rate (CCR), age at first calving (AFC), heifer conception rate (HCR), daughter calving ease (DCE), daughter still birth (DSB), and daughter pregnancy rate (DPR). For comparison, we also randomly shifted RNA editing sites for the permutation test with blue color. Error bars represent the standard deviation of permutation tests. (I) The chromatogram of DNA and cDNA in downstream of *TMED10* (chr10:86373104-86373883) by Sanger sequencing. The validated editing sites were marked with yellow background.

mammary. Most of the common editing sites had low levels in the non-lactating mammary gland but moderate and high levels in lactating mammary gland (Fig 5F). We identified 104 RNA editing sites with different editing levels between lactation (adjusted P-value < 0.05, Fig 5G). Most differential RNA editing sites were up-regulated in lactation. These genes close to differential RNA editing sites were associated with immune response, RNA phosphodiester bond hydrolysis, oxygen transporter activity, secreted, and *Staphylococcus aureus* infection (Fig 5D and S8 Table). Interestingly, we observed three differential RNA editing sites located in the second intron of *CSN3*. To assess the contribution of genetic variance by RNA editing, we captured proximal SNPs within RNA editing sites and their flanking 100 Kb regions (Table 1). We also randomly shifted RNA editing sites for the permutation test for comparison. We detected that RNA editing sites in both two stages union and stage-specific or different groups could explain large genetic variance than randomly shifted sites for milk production traits (Fig 5H). RNA editing captured more genetic variance for MFP and MPP compared to other milk production traits.

## Histone modification, DNA methylation, eQTLs, and sQTLs

We assayed the four histone modifications (H3K4Me1, H3K4Me3, H3K27Me3 and H3K27ac) and one transcription factor (CTCF) across six tissues. By linking regulatory regions to genetic variants, we detected the regulatory regions captured a greater amount of genetic variance for MFP and MPP (Fig 6A). These active promoters or enhancers (H3K27ac, H3K4Me1, and H3K4Me3) could explain relatively more genetic variance for all five milk production traits (Table 1) than the other two repressive regulatory elements (CTCF and H3K27Me3). All five milk production traits showed greater genetic variance in the mammary and liver compared to other tissues for these active promoters or enhancers. However, the results for repressive regulatory elements were the opposite (Fig 6A).

We identified 49,914,904, 48,794,561, and 33,975,104 DNA methylation sites in mammary gland, whole blood cells (WBC), and brain, respectively, which could be merged into 241,817, 208,584, and 136,113 DNA methylated regions. We found strong evidence for mammary and blood, and less so for the brain, that DNA methylated regions with small changes of methylation levels between lactation stages (|Δlevel|≤0.2) explained more genetic variance than that with large changes (|Δlevel|>0.2, Fig 6B). The mammary gland and blood methylation could capture more genetic variance of MFP than that of other traits (S16 Fig). After adjusting for SNP proportion, the odds ratios for genetic variance explained by mammary gland methylation regions were significantly larger than those for blood methylation regions, as shown in Fig 6C. We identified 5,525, 8,191, and 3,026 differentially methylated regions (DMRs, $P < 1 \times 10^{-5}$) between lactation and non-lactation in mammary, WBC and brain, respectively. These DMRs had a more significant difference in odds ratios between mammary gland and blood compared to all methylation regions (Fig 6C). We then classified variants into two classes based on whether expression or splicing quantitative trait loci (eQTL or sQTL). The eQTL and sQTL explained 0.07~0.32 and 0.12~0.44 variance proportion for milk production traits, respectively (Table 1). The genetic variance explained by the eQTLs was higher than expected when we shifted eQTLs to two new positions as permutations (Fig 6D). Both eQTL and sQTL could capture large genetic variances for MFP and small genetic variance for MPY.

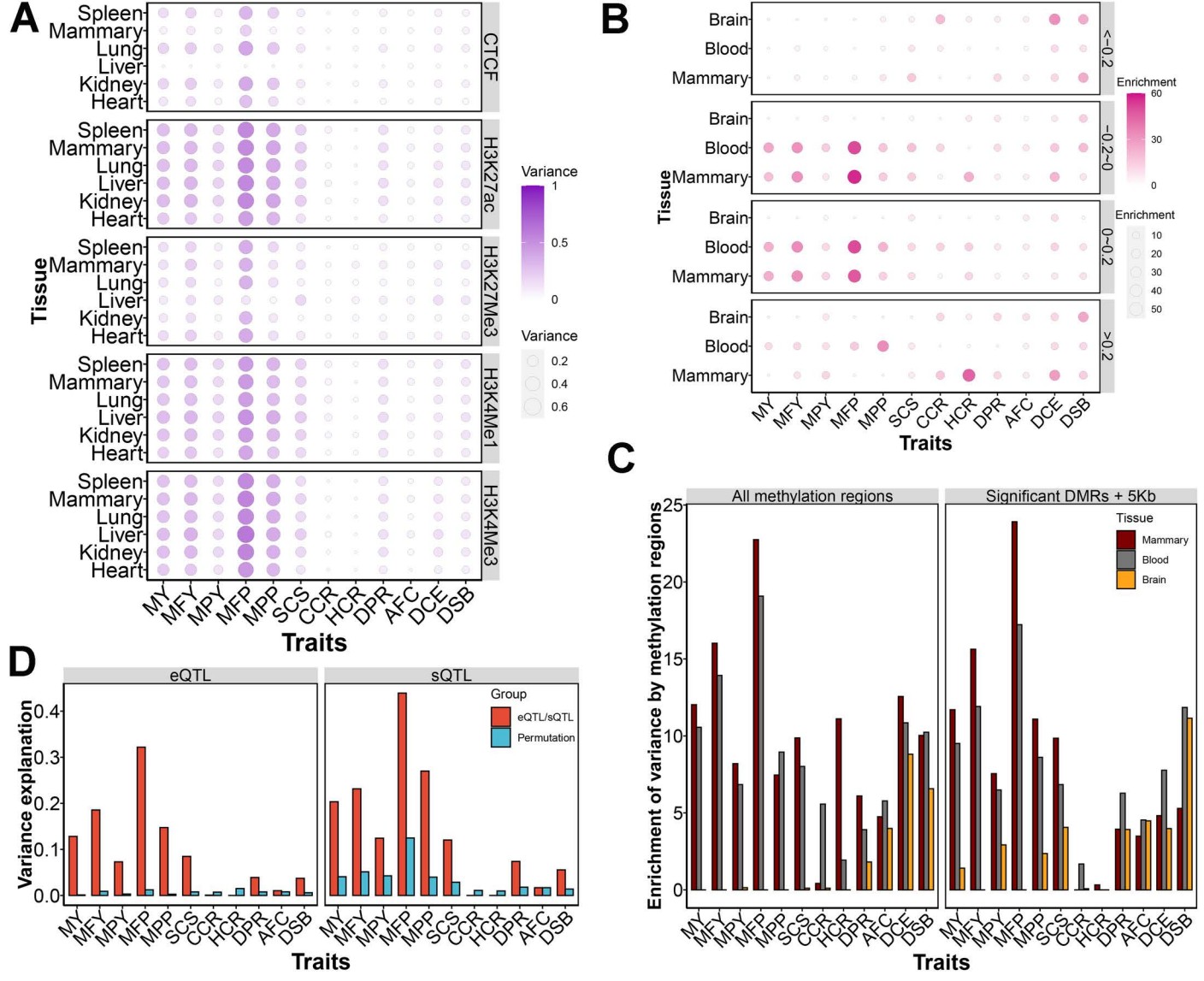

**Fig 6. The proportion of variance explained by histone modification, DNA methylation, mammary eQTL and mammary sQTL.** (A) The estimation of the variance explained by histone modification in six tissues for milk yield (MY), milk fat yield (MFY), milk protein yield (MPY), milk fat percentage (MFP), milk protein percentage (MPP), somatic cell score (SCS), cow conception rate (CCR), age at first calving (AFC), heifer conception rate (HCR), daughter calving ease (DCE), daughter still birth (DSB), and daughter pregnancy rate (DPR). (B) The enrichment of genetic variance is explained by DNA methylation with different changes of methylation levels between lactation and non-lactaing period in mammary, blood and brain. The enrichments were calculated using the odd ratios between the proportion of variance explanation and the proportion of SNP number. (C) The enrichment of genetic variance is explained by differential DNA methylation regions (DMRs) between lactation and non-lactaing period in mammary, blood and brain. (D) The estimation of the variance is explained by eQTLs and sQTLs of lactating mammary. For comparison, we also randomly shifted eQTL or sQTL for the permutation test with blue color.

## Genome prediction of milk production traits using prior information

We incorporated GRMs for each functional class into a REML model, which allowed us to estimate the genetic variance component contributed by each functional class simultaneously. Fig 7A and Table 2 illustrate the proportions of variance captured by each functional class of SNPs. We detected non-functional SNPs, sQTLs, and histone modification, and

DEGs could explain the largest genetic variance in most traits. After we corrected the SNP number in each functional class, we observed that the sQTL had the highest genetic variance enrichment for all traits except for MFY, which was the most significantly enriched by eQTLs (Fig 7B). The DEGs, histone, miRNAs, and RNA editing had a modest genetic variance enrichment for most traits. Surprisingly, the DNA methylation explained almost no genetic variance.

In theory, sequencing variants could help improve the accuracy of genomic selection. However, utilizing them in genomic prediction with millions of animals is impractical due to computational constraints. A compromise approach is to use partially effective SNPs, which could be selected from our functional classes. We merged 979,240 SNPs into the 671K genotype panel, commonly used in genomic evaluations of US Holstein bulls. Then we pruned them based on LD

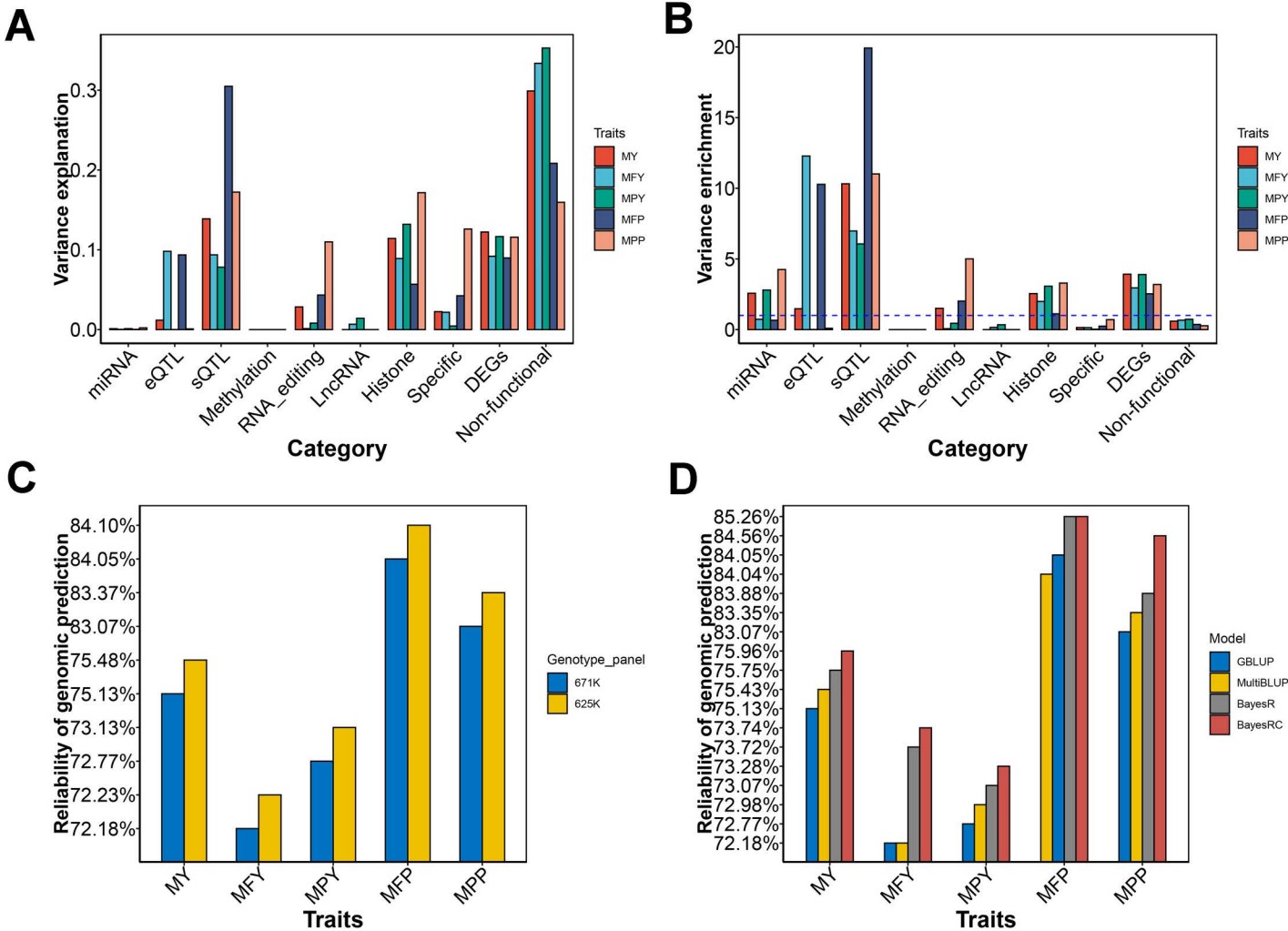

**Fig 7. Mammary omic data enhance understanding of the genetic architecture of milk production traits and genomic predictions.** (A) The proportion of the variance captured by each functional class of SNPs when fitting the genomic relationship matrix for each functional class simultaneously in the REML model. (B) The enrichment of genetic variance is explained by each functional class. The enrichments were calculated using the odd ratios between the proportion of variance explanation and the proportion of SNP number. (C) The predictive reliability of milk production traits by 671K and 625K genotype panel. The 671K is a genotype panel for routine evaluations of dairy bulls. 625K were our new refined genotype panel, containing 323K old variants and 302K new variants extracted from imputed sequence variants based on functional classes. (D) The comparison of predictive reliability of milk production traits between GBLUP and multiBLUP model, as well as between BayesR and BayesRC model.

thresholds ($r^2 < 0.9$) and generated a new genotype panel with 624,759 tagged SNPs, which included 322,981 original SNPs and 301,778 functional SNPs. We found that the genetic variation explained using the new genotype panel was improved by 1.7% compared to the original genotype panel (S17 Fig). Using the GBLUP model, the new genotype panel achieved higher reliability of genomic selection by 0.22% than the original genotype panel (Table 3 and Fig 7C). The reliability increment obtained using the functional genotype panel was more noticeable in MY and MPY.

Another strategy for utilizing prior biological information involves genotype partitioning. We divided the original 671K genotype panel into two groups based on whether the SNPs were in functional classes. We fitted these two SNP sets simultaneously in the REML or BayesRC models to conduct two-component genomic predictions. Compared to the GBLUP model, prior biological information resulted in higher predictive reliability with MultiBLUP for MY, MPY, and MPP (average increase of 0.15%, Table 4 and Fig 7D). Compared to the BayesR model, prior biological information resulted in higher predictive reliability with BayesRC for MY, MPY, MFP, and MPP (average increase of 0.21%, Table 4 and Fig 7D). The reliability increment obtained using bayesRC with prior biological information was more noticeable for MPP with a 0.68% increment. Prior biological information did not achieve an increase > 0.1% in predictive reliability for MFY and MFP in both MultiBLUP and BayesRC models.

## Discussion

In this study, we analyzed several sources of external information based on mammary gland omics data. Using a statistical model, we assessed the genetic variance explained by each class of variants when integrated with a large population

**Table 2. The proportion of variance explained for each functional class across five milk production traits using ten genomic relationship matrices (GRMs) model, which simultaneously fitting all nine functional classes (DEGs, lncRNAs, miRNAs, RNA editing, DNA methylation, histone modifications, eQTLs, sQTLs, splicing variants) along with one non-functional class and their respective GRMs.**

| Functional class | Variants number | MY | MFY | MPY | MFP | MPP |
|---|---|---|---|---|---|---|
| Specific genes ± 5Kb | 631,689 | 0.022 | 0.022 | 0.004 | 0.042 | 0.126 |
| DEGs ± 5Kb | 127,947 | 0.122 | 0.092 | 0.117 | 0.090 | 0.116 |
| DE lncRNAs ± 5Kb | 177,123 | 0.000 | 0.007 | 0.014 | 0.000 | 0.000 |
| DE miRNAs ± 5Kb | 1,796 | 0.001 | 0.000 | 0.001 | 0.000 | 0.002 |
| RNA editing ± 100Kb | 77,442 | 0.028 | 0.001 | 0.008 | 0.043 | 0.110 |
| DMRs | 61,661 | 0.000 | 0.000 | 0.000 | 0.000 | 0.000 |
| Enhancers | 183,768 | 0.114 | 0.089 | 0.132 | 0.057 | 0.172 |
| eQTLs | 32,830 | 0.012 | 0.098 | 0.000 | 0.094 | 0.001 |
| sQTLs | 55,166 | 0.139 | 0.094 | 0.078 | 0.305 | 0.172 |
| Other variants | 2,062,309 | 0.299 | 0.334 | 0.353 | 0.208 | 0.160 |
| Total | 3,026,716[a] | 0.738[b] | 0.736[b] | 0.707[b] | 0.839[b] | 0.858[b] |

[a]Total item refers to the number of all genotype variants.

[b]The total proportion of variance for each trait is the sum of the proportions of variance from all functional classes. All abbrevations are same with above.

**Table 3. The reliability of genomic prediction for five milk production traits using two different genotype panel based on GBLUP.**

| Genotype panel | MY | MFY | MPY | MFP | MPP |
|---|---|---|---|---|---|
| 671K | 75.13% | 72.18% | 72.77% | 84.05% | 83.07% |
| 625K (Old 323K + New 302K) | 75.48% | 72.23% | 73.13% | 84.10% | 83.37% |
| Increase | 0.35% | 0.05% | 0.36% | 0.04% | 0.29% |

The 671K is a genotype panel for routine evaluations of dairy bulls. 625K were our new refined genotype panel, containing 323K old variants and 302K new variants extracted from imputed sequence variants based on functional classes. Five milk production traits include milk yield (MY), milk fat yield (MFY), milk protein yield (MPY), milk fat percentage (MFP) and milk protein percentage (MPP).

**Table 4. The reliability of genomic prediction for five milk production traits based on GBLUP, MultiBLUP, BayesR, and BayesRC.**

| Traits | MY | MFY | MPY | MFP | MPP |
|---|---|---|---|---|---|
| GBLUP | 75.13% | 72.18% | 72.77% | 84.05% | 83.07% |
| MultiBLUP | 75.43% | 72.18% | 72.98% | 84.04% | 83.35% |
| Increase | 0.30% | 0.00% | 0.20% | -0.01% | 0.28% |
| BayesR | 75.75% | 73.72% | 73.07% | 85.26% | 83.88% |
| BayesRC | 75.96% | 73.74% | 73.28% | 85.26% | 84.56% |
| Increase | 0.21% | 0.02% | 0.22% | 0.00% | 0.68% |

Five milk production traits include milk yield (MY), milk fat yield (MFY), milk protein yield (MPY), milk fat percentage (MFP) and milk protein percentage (MPP).

of genotypes and phenotypes. Other researchers can leverage this additional information to annotate their own variants of interest. Our present research approach aims to provide innovative information about the genetic and biological mechanisms that support milk production traits. The genomic prediction reliability of milk production traits was improved when using these functional classes, which will strengthen genomic improvement programs for dairy cattle.

The phenotype data were derived from deregressed PTAs, which were generally highly reliable due to the presence of many phenotyped daughters for each bull. The estimation of heritability using deregressed PTAs are usually higher than when using original phenotype [23]. We estimate that 71.1%～89.2% of phenotypic variation for milk production traits is tagged by autosomal SNPs. The chromosome segments explain variation in approximate proportion to their lengths, while the linear relationship between the estimate of genetic variance explained and genomic size is not perfect, especially for MFP and MPP. Chromosome 14 captured a large genetic variance for most milk production traits, which is attributed to the major locus *DGAT1* on this chromosome that control milk fat and yield traits [24–27]. The *GHR* is another major locus for milk production traits located in chromosome 20 [28,29]. The more expected genetic variances of MPP, MFP, and MY could be explained by chromosome 20. The four caseins (αS1-, αS2-, β- and κ-CN) account for almost 80% of the whole bovine milk protein [30]. The genetic variance of MPP was largely captured by chromosome 6, which carries the *ABCG2* and casein genes that are known to affect protein percentage [31–33]. Chromosome 5 could account for a large genetic variance for all five milk production traits, which may be attributed to the presences of milk-related causative genes, such as *LALBA* [34], *CSF2RB* [35], and *MGST1* [36] on this chromosome. For all five milk production traits, the 5'-UTR, splicing regions, synonymous and 3'-UTR classes explained more variation than expected by chance, which implied the causative variants were enriched in these regulatory and exonic regions.

As fat-filled adipocytes comprise a large proportion of the stromal fat pad in the non-lactating mammary gland [37], the mammary parenchyma and mammary fat pad are closer to the non-lactating mammary gland. Milk cell transcriptome represents the lactating mammary glands and can be used as effective and alternative samples to study mammary gland gene expression [38]. Milk cell is moderately correlated with lactating mammary gland based on their gene specificity value (t-statistics), implying that somatic milk cells can serve as an indicator for studying gene expression in lactating mammary tissue but cannot fully replace it. Moreover, the specifically expressed genes in the lactating mammary gland capture more genetic variances of milk production traits than those in the other four mammary tissues. The specifically expressed genes in the liver captured considerable genetic variances for milk production traits, indicating its active and complex functions in synthesizing proteins and fat.

One-third of the genes expressed in the mammary gland are differentially expressed during lactation, suggesting that lactation is a complex process that requires the participation of multiple genes. Many of the up-regulated genes were found to be involved in metabolic pathways related to milk fat, milk protein, lactose, and oxytocin, which could promote lactation and the synthesis of milk components. In contrast, the down-regulated genes were mainly involved in immunity, disease resistance, and repair processes, indicating that recovery is essential for the non-lactating period. Genetic

variance partitioning demonstrated that the up-regulated genes contributed more to the heritability of milk production traits than the down-regulated genes. These findings suggest that the significantly up-regulated DEGs harbor variants that can play a role in trait variation and therefore warrant further investigation. LncRNAs might be important regulators for the lactation cycle [39,40] and related to the synthesis of milk protein [9] and fat [41]. In this study, thousands of lncRNAs are differentially expressed during lactation, these down-regulated lncRNAs in lactation capture unneglectable genetic variance for milk production traits.

The genetic variance explained per variant was greater for DE miRNAs than all miRNAs in most milk production traits, confirming that these DE miRNAs are likely to be involved in lactation-related activities. As the simple target prediction algorithms of miRNA may generate a large portion of false positive miRNA targets, it is important to use complementary approaches that integrate expression data to enhance the accuracy of predicted miRNA-mRNA interactions. By integrating the information on miRNA-mRNA targeting and their expression correlation, we demonstrate that the expression of miRNAs can be associated with the negative expression of a subset of predicted target mRNAs in mammary glands, leading to a more focused set of miRNAs to validate functionally. Nearly two-thirds of the 53 potentially functional miRNAs are differentially expressed between non-lactating and lactating periods, suggesting that these miRNAs are more likely to be involved in lactation-related processes. The genomic partitioning proved that up-regulated miRNA precursors and their targets contained variants that captured larger genetic variance than down-regulated miRNAs. Our study introduces a strategy for identifying potential miRNAs that are associated with a particular trait. This approach involves cross-referencing the significant enrichment of targets-correlation with a target's relatively high genetic variance explained, thereby identifying overlapping results that may suggest candidate miRNAs for a trait.

Although millions of RNA editing sites have been reported in cattle, RNA editing events related to lactation have not yet been investigated. Consistent with previous studies, most RNA editing sites were located in non-coding repetitive regions of the transcriptome [42,43]. The expressions of *ADAR* were significantly down-regulated in lactation. Meanwhile, the RNA editing numbers, events, and levels were lower in lactation than in the non-lactating period, indicating that RNA editing activity was inhibited during mammary lactation. Adenosine deamination is a prominent form of RNA editing in the mammary gland transcriptome, with over 98% of editing sites being of the A-to-G type. Previous studies on the bovine genome have reported that RNA editing events in protein-coding regions are rare [44,45]. Similarly, our analysis confirms that RNA editing occurrences in protein-coding regions are infrequent in cattle mammary gland. Thousands of RNA editing sites occurred in all six non-lactating or lactating mammary gland samples, suggesting these editing sites could be widespread in specific stages of lactation. Certain RNA editing sites that occur repeatedly across different stages may have important biological functions. The specifically edited genes in lactating mammary gland are mainly associated with metabolic processes, and protein transportation and secretion. In contrast, these specifically edited genes in non-lactating mammary gland were primarily related to immune activity. The differentially edited genes were associated with both immune response and metabolic process. RNA editing sites in both two stages union and stage-specific or different groups could explain large genetic variance than randomly shifted sites for milk production traits, especially for MFP and MPP, suggesting that RNA editing could play an essential role in regulating lactation activity.

Histone modifications such as H3K27ac, H3K4Me1, and H3K4Me3 correlated with active regions of the genome [46,47], while CTCF and H3K27Me3 have been found to represent repression of transcriptional activity [47,48]. Consistent with the functions of regulatory elements, active promoters or enhancers explain more genetic variance for milk production traits than repressive regulatory elements. The regulatory regions captured more genetic variance for MFP than other milk production traits, which implied the causative variants of MFP are largely located in these regulatory regions. DNA methylation regions in mammary gland and blood with minimal alterations in methylation levels between lactation stages exhibited a higher degree of variation explained than those with substantial changes, which implied causative variants are relatively conservative and difficult to undergo major changes. The DNA methylation regions in the mammary gland showed a higher enrichment of genetic variance for milk production traits than the other two tissues, suggesting that the

mammary gland is the primary tissue for investigating the genetic mechanisms underlying milk production traits. The eQTLs and sQTLs could explain considerable genetic variation for milk production traits, especially for MFP, consistent with a previous study [49]. When correcting the SNP number in each functional class, the sQTLs and eQTLs had the larger genetic variance enrichment for most traits. This proves these classes contain variants that can contribute to trait variation and should be prioritized in further studies.

To effectively handle the vast number of genotyped animals, diverse data sources, and millions of variants, it is crucial to employ computational strategies that efficiently optimize the balance between imputation, selection, and prediction costs. Previous studies have reported that adding efficient sequence variants could improve the reliabilities of genomic predictions [50,51]. We added the pre-selected variants in functional classes and removed redundant SNPs based on LD. The new SNP sets could improve the reliabilities of genomic predictions by 0.22%, which has the potential to apply to the genomic selection of milk production traits and accelerate the genetic improvement of dairy cattle. As the genomic variants of milk production traits seem to be enriched in certain genome regions, the assumption of the GBLUP approach that *a priori* all markers contribute equally to trait variability does not hold well. The genomic variants in these enriched regions have greater weights than the remaining variants in MultiBLUP and BayesRC [18,20,52]. The MultiBLUP and BayesRC models based on functional classes have greater increases in predictive reliability compared with those in MY, MPP, and MPY, thus reflecting that our functional classes have the potential to accelerate the genetic improvement of these traits. Due to the complex LD of major QTLs (such as *DGAT1*) in milk fat [53], the increase of predictive reliability with Multi-BLUP and BayesRC was limited in MFY and MFP. More accurate information about the causal genomic variants should be refined for genomic prediction of MFY and MFP traits.

## Conclusions

In summary, the findings emphasize that incorporating mammary gland biological priors enhance our understanding of phenotypic diversity's genetic basis. The significant genetic variance explanation provided by various functional classes indicates that these classes contain variants capable of contributing to trait variation, highlighting the importance of prioritizing them in future studies. The proposed candidate miRNAs and RNA editing sites may contribute to future applications in molecular-assisted breeding. Compared to GBLUP and BayesR, MultiBLUP and BayesRC models increased the reliability of genomic prediction for milk production traits in dairy cattle by incorporating biological information of multiple omic data from the mammary gland, thus providing novel biological insights into the genetic basis of milk production traits.

## Materials and methods

### RNA sequencing data

We uniformly analysed 6,642 RNA-seq data sets from Sequence Read Archive (SRA, https://www.ncbi.nlm.nih.gov/sra/) and CNCB databases (https://ngdc.cncb.ac.cn/) to calculate gene expression (S9 Table). Trimmomatic (v.0.39) was utilized to remove adapters and eliminate low-quality reads [54]. STAR aligner (v.2.7.0) was employed to map the clean reads to the reference genome of Bos taurus (ARS-UCD1.2) [55]. Gene expression was calculated by FPKM based on a mapped file using Stringtie (v.2.1.1) [56]. To investigate the landscape and dynamic changes of genes during lactation in dairy cattle, we extracted the raw read counts of 103 biopsies mammary RNA-seq samples using featureCounts (v.1.5.2) [57], including 42 lactating and 61 non-lactating animals (S2 Table). We conducted the differential analysis for gene expression using DEseq2 with the condition: Project+Stage [58]. Genes were considered differentially expressed if they had an adjusted P-value below 0.05. To identify the novel lncRNA, we first built the novel transcripts using Stringtie (v.2.1.1) [56] and were guided by the Ensembl gene models of gffcompare [59]. We extracted transcripts that shared the same start position, end position, and exon-intron boundary, and were supported by at least five samples. Transcripts with length ≥ 200nt, exon ≥ 2, and maximum length of open reading frame (ORF) less than 120 amino acids (360 bp), as well as annotated by 'u', and 'i', were obtained. We predicted the protein-coding potential for each candidate transcript using

CPC2 [60], PLEK [61], and CNCI [62]. Transcripts with protein-coding potential score > 0 in either software were removed. To obtain amino acid sequences, all transcripts were translated across all three reading frames. To eliminate transcripts containing known protein domains, we conducted a search against the Pfam database (version 30.0) and subsequently removed all matching transcripts [63].

## Mammary-specific gene expression

We classified 152 tissues or cell types into 13 tissue categories based on established biological knowledge (S9 Table) [64]. When we calculated the t-statistic of each gene for a tissue, all samples from a same tissue category were excluded. For example, when computing the *t*-statistic of each gene in the lactating mammary, we compared expression in lactating mammary samples to expression in non-mammary tissues, excluding non-lactating mammary, mammary fat pad, mammary parenchyma, and milk cells. Thus, for each gene, the null hypothesis is that there is no significant difference in the gene expression level between lactating mammary tissue and non-mammary tissues. The *t*-statistics were calculated using a general linear model:

$$Y = Xb + e \tag{1}$$

where $Y$ represents gene expression that has been log2-transformed and scaled using Z-score normalization within each tissue. $X$ was a design matrix, with each row corresponding to a sample. The first column of $X$ had a '1' for every lactating mammary sample and a '–1' for every non-mammary sample. The remaining columns were intercept and covariates (i.e., project, breed, sex, and age). $b$ is the corresponding tissue effect. $e$ was the residual effect. We calculated *t*-statistics using ordinary least-squares according to Finucane's formula [65].

## MicroRNA sequencing data

Samples from 12 small RNA datasets (SRA with accession number PRJNA689373) were collected from six cows at approximately 79 days postpartum (i.e., the peak lactation period) and from another six cows during the non-lactating period, as described in our previous study [3]. Cutadapt (v.4.4) and Trimmomatic (v.0.39) were employed for quality trimming and adaptor removal of the Illumina reads [54,66]. We utilized miRDeep2 to map the cleaned reads, ranging from 18 nt to 30 nt in length, to the Bos taurus reference genome (ARS-UCD1.2) [67]. To investigate differentially expressed miRNAs between lactation and non-lactating periods, read counts were modeled using a generalized linear model, taking into account the experimental design with lactation stages (lactation and non-lactation) using the DESeq2 R package [58]. MicroRNAs were considered differentially expressed if they had an adjusted P-value below 0.05.

The potential miRNA targets were predicted using miRanda (v.1.0b) with the default parameters [68]. Additionally, the Pearson correlation coefficients between the specific miRNA and its predicted target mRNAs were calculated. For each miRNA, a 2 × 2 contingency table was constructed for all mRNAs, categorizing them based on whether they exhibited a negative correlation with the target miRNA (correlation < 0 and P-value of correlation ≤ 0.05) and whether they were predicted targets of the miRNA. This table was then utilized to assess the enrichment level of negatively correlated mRNAs among the predicted targets of the intended miRNA using Fisher's exact test. If the P-value derived from Fisher's exact test was found to be less than 0.05, the miRNA was considered to have a significant number of mRNA targets with a negative correlation. As a result, it was selected as a significant miRNA in the screening process.

## RNA editing identification

For identifying RNA editing sites in RNA-seq data alone, we employed a clustering strategy to align and examine the unmapped reads [69]. Firstly, we extracted all unmapped reads from the initial alignment carried out using STAR aligner with a mismatch threshold set at > 3. Next, we transformed all "A" nucleotides to "G" nucleotides in both the unmapped

reads and the reference genome. The transformed RNA reads were then realigned to the transformed reference genome using BWA (v.0.7.17) [70]. The resulting mapped reads were converted back to their original sequences, which were considered as candidate RNA editing reads. To enhance the accuracy of identifying RNA editing clusters, we established specific criteria. Specifically, we considered the number of A-to-G mismatches that constituted at least 5% of the read length (or at least three A-to-G mismatches for read lengths ≤60 bp) and accounted for more than 80% of the total number of mismatches. To avoid false positives stemming from technical artifacts, additional filters were applied, including requiring average Phred quality scores to be >25 and removing reads containing >10% ambivalent nucleotides, >10 simple repeats, or >20 successive single nucleotides. This procedure was repeated for the other 11 types of editing events (e.g., A-to-C and G-to-A). The sequencing library was not stranded, so the A-to-G edited sites may appear as T-to-C mismatches, but T-to-C editing rarely occurs in bovines, so all T-to-C sites were treated as A-to-G sites. The regions of RNA editing clusters were defined as the segment of the edited read starting from the first A-to-G mismatch and ending at the last A-to-G mismatch, with a distance ≤100 bp between them. To calculate RNA editing levels, we considered both mapped and unmapped reads. Additionally, the STAR-mapped alignments were improved using Picard tools. We extracted the depth of each RNA editing site using the REDItools [67]. Then, we computed the RNA editing level for a given site as described below:

$$\text{RNA editing level} = \frac{\text{mapped G } + \text{ unmapped G}}{\text{mapped depth } + \text{ unmapped G}}$$

(2)

### Histone modification and DNA methylation

Four histone modifications (H3K4Me1, H3K4Me3, H3K27ac, and H3K27Me3) and one transcription factor (CTCF) in 6 tissues (heart, kidney, liver, lung, mammary, and spleen) were collected from two or three lactating Holstein dairy cows, depending on the tissue. Details were reported in Prowse-Wilkins et al. [71] and data may be retrieved from SRA with accession number PRJEB41939. Trimmomatic (v.0.39) was utilized to remove adapters and eliminate low-quality reads [54]. BWA (v.0.7.17) was employed to map the clean reads to the reference genome of Bos taurus (ARS-UCD1.2) [70]. Uniquely mapped reads were identified using SAMtools [72]. Both ChIP and input reads were used to call peaks using MACS2 [73]. The peaks with biological replicates were combined and returned a union of all peak locations by ChIP-R [74]. The DNA methylation data of mammary glands, whole blood cells, and prefrontal cortex of the brain were collected from one lactating and one non-lactating cow, which has been reported in a previous study with SRA accession number GSE106538. Bismark (v.0.14.5) was employed to map the clean reads to the reference genome of Bos taurus (ARS-UCD1.2) [75]. DMRs were identified by metilene (v.0.2.8) with P-value $<1 \times 10^{-5}$. The cis-eQTL and cis-sQTL data of lactating mammary were obtained from Cattle GTEx.

### Variance component analysis

All the 23,566 Holstein bulls used in this study had highly reliable PTAs for 12 production and reproduction traits, which have been discribed in previous studies [13,76]. The PTAs represent breeding values after removing fixed non-genetic effects, and their reliabilities were used to quantify the amount of information available for different individuals [77]. De-regressed PTAs were calculated as described by Garrick et al., by dividing the PTA by its squared reliability, thereby excluding parental information and decreasing inter-animal dependency [78]. These de-regressed PTAs were then used as the phenotype in subsequent analyses. Genotype data included SNP and insertion-deletion (InDel) calls from 1000 Bull Genomes Project, described in detail previously [50,76]. We lifted the sequence variants to version ARS-UCD1.2 of the *Bos taurus* genome assembly using liftOver. SNPs with minor allele frequencies <0.05, genotype call rates below 90%, located in non-autosomal regions, and showing significant Hardy-Weinberg disequilibrium at $1 \times 10^{-6}$, along with samples exhibiting call rates less than

90%, were excluded from subsequent analysis by PLINK v1.90 [79]. After quality control, 3,026,716 autosome variants were available for variance component analysis. To partition the total genetic variance onto individual chromosomes, sequence variants were split into 29 components, one for each autosome in the cattle genome. Sequence variants were annotated into intergenic, intron, missense, synonymous, 5'UTR, 3'UTR, 5 Kb upstream, 5 Kb downstream, splicing region variants with Ensembl Variant Effect Predictor (VEP) [80]. To calculate the genetic variance explained by all genotype variants, we employed restricted maximum likelihood (REML) using GCTA software with the following mode [81]:

$$\mathbf{y} = \mathbf{1}\mu + \mathbf{g} + \varepsilon \tag{3}$$

where $\mathbf{y}$ is a vector of deregressed PTAs; $\mathbf{1}\mu$ is a vector of trait means. $\mathbf{g}$ is a vector of total additive genetic effects $\sim N$ $(\mathbf{0}, \mathbf{G}\sigma_g^2)$; where $\mathbf{G}$ is the genomic relationship matrix (GRM) and $\sigma_g^2$ is the additive genetic variance; $\varepsilon$ denotes random residual errors $\sim \boldsymbol{N}$ $(\mathbf{0}, \sigma_\varepsilon^2)$, where $\sigma_\varepsilon^2$ is the error variance. $\sigma_P^2$ is the phenotypic variance. The heritability ($h^2 = \sigma_g^2 / \sigma_P^2$) was the proportion of phenotypic variance ($\sigma_P^2$) explained by all variants together.

To perform variance component analysis for each functional class, we first constructed the GRM from the SNPs within each functional class ($\boldsymbol{G_f}$). We then estimated the genetic variance attributable to each functional class by fitting the GRMs of all the functional classes simultaneously in the model:

$$\mathbf{y} = \mathbf{1}\mu + \sum_{f=1}^{N} \boldsymbol{g_f} + \varepsilon \tag{4}$$

where $\boldsymbol{g_f}$ is a vector of genetic effects attributable to each functional class and Var ($\boldsymbol{g_f}$) = $\boldsymbol{G_f}\sigma_f^2$; the proportion of variance explained by each functional class is defined as $h_f^2 = \sigma_f^2 / \sigma_P^2$; $N$ is the number of classes. The two GRMs model was derived from Model 4 when $N = 2$, analyzing each functional class (e.g., DEGs, miRNAs) separately against the non-functional SNP background and their respective GRMs. The ten GRMs model was derived from Model 4 when $N = 10$, by simultaneously fitting all nine functional classes (DEGs, lncRNAs, miRNAs, RNA editing, DNA methylation, histone modifications, eQTLs, sQTLs, splicing variants) along with one non-functional class and their respective GRMs. The variance proportion ($h_f^2$) for each class was calculated as the ratio of the class variance to the total variance. To correct for bias caused by different numbers of SNPs in each class, we computed the enrichment value using the odd ratios between the proportion of variance explained and the proportion of SNP number:

$$Enrichments = \frac{h_f^2 / h_{all}^2}{n_f / n} \tag{5}$$

where $h_{all}^2$ denotes the proportion of phenotypic variance explained by the total sum of variance from all functional classes combined; $n$ is the total number of variants; $n_f$ is the number of variants found in the functional class. We estimated whether SNPs within functional classes, such as RNA editing sites, eQTLs, and sQTLs, explain a greater amount of variance compared to randomly selected variants from a shifted permutation test with an equivalent number of variants. To perform the shifted permutation test, we first converted the genomic position of variants in the functional class into a continuous position. For a variant with genomic position B in chromosome N, its continuous position should be $\sum_{i=1}^{N-1} l_i + B$, where $l_i$ denotes the length of chromosome i. Then the observed variants set was shifted to the new continuous ($P_1, P_2, P_3, \ldots, P_n$) based on random value R within $1 \sim \sum_{i=1}^{29} l_i$ using the following formula:

$$P_i = \begin{cases} \sum_{i=1}^{N-1} l_i + B + R, & N_i + R \leq \sum_{i=1}^{29} l_i \\ \sum_{i=1}^{N-1} l_i + B + R - \sum_{i=1}^{29} l_i, & N_i + R > \sum_{i=1}^{29} l_i \end{cases} \tag{6}$$

The new positions were recovered into genomic positions based on autosome length. This shifted permutation test using Perl scripts is available in an online code repository (https://github.com/WentaoCai/Permutation).

## Genomic prediction

The Holstein cattle population was divided into a reference population (n = 19,575) and a validation population (n = 3,991) based on their year of birth [50]. The genotype with 671,193 autosome variants were commonly used in US Holstein bulls' genomic selection. We used two strategies to optimize genomic prediction for milk production traits. Firstly, we merged 979,240 functional SNPs into the 671K genotype panel and pruned them based on LD thresholds ($r^2 < 0.9$) and generated a new genotype panel with 624,759 tagged SNPs 322,981 original SNPs and 301,778 functional SNPs. Secondly, we divided the original 671K genotype panel into two groups based on whether the SNPs were in functional classes. We simultaneously fitted these two SNP sets in the REML or BayesRC models to conduct two-component genomic predictions.

## The GBLUP model was based on a linear mixed model

$$\boldsymbol{y} = \boldsymbol{1}\mu + \boldsymbol{Zg} + \boldsymbol{e} \tag{7}$$

where $\boldsymbol{y}$ denotes a vector of the deregressed PTAs obtained from phenotypic records; $\mu$ is the overall mean; $\boldsymbol{Z}$ is a design matrix that allocates phenotypic records to individuals; $\boldsymbol{g}$ is a vector of additive genetic values and assumed that $\boldsymbol{g} \sim \boldsymbol{N}(0, \boldsymbol{G}\sigma_g^2)$, in which $\boldsymbol{G}$ is the GRM constructed based on the genomic marker information [82].

The MultiBLUP model was based on a linear mixed model, including multiple random genomic effects. Here, we used two random genomic effects:

$$\boldsymbol{y} = \boldsymbol{1}\mu + \boldsymbol{Zg_f} + \boldsymbol{Zg_r} + \boldsymbol{e} \tag{8}$$

where $\boldsymbol{y}$, $\boldsymbol{Z}$, and $\boldsymbol{e}$ are the same as terms described in the GBLUP model; $\boldsymbol{g_f}$ is the vector of genomic values captured by genetic markers linked to the functional classes; $\boldsymbol{g_r}$ is the vector of genomic values captured by the remaining set of genetic markers. The random genetic effects are assumed to be independent normally distributed values $\boldsymbol{g_f} \sim N(0, \boldsymbol{G_f}\sigma_f^2)$, $\boldsymbol{g_r} \sim N(0, \boldsymbol{G_r}\sigma_r^2)$. $\boldsymbol{G_f}$ and $\boldsymbol{G_r}$ are the GRM constructed based on the genomic marker within and outside functional classes, respectively. Both GBLUP and MultiBLUP models were implemented using LDAK v5.2 software [21].

We performed BayesR and BayesRC using the following model:

$$\boldsymbol{y} = \boldsymbol{1}\mu + \boldsymbol{Ws} + \boldsymbol{e} \tag{9}$$

where $\boldsymbol{y}$ and $\boldsymbol{e}$ are the same as terms described in the GBLUP model, $\boldsymbol{W}$ is a design matrix that allocates phenotypic records to individuals, $\mathbf{s}$ is the sum of the vector of SNP effects derived from different assumed distributions. BayesR assumes that SNP effects follow a mixture of four normal distributions $N(0, \gamma_k\sigma_k^2)$, the $\gamma_k$ are 0, 0.01, 0.1 and 1 with probability $\pi_1$, $\pi_2$, $\pi_3$ and $\pi_4$, respectively, and $\pi_1 + \pi_2 + \pi_3 + \pi_4 = 1$ [83]. The variant effects for members within functional classes are assumed to belong to a mixture of four normal distributions with proportions ($\pi_{f1}$, $\pi_{f2}$, $\pi_{f3}$ and $\pi_{f4}$) while the variant effects that are members outside functional classes belong to an independent mixture of the four distributions with proportions ($\pi_{r1}$, $\pi_{r2}$, $\pi_{r3}$ and $\pi_{r4}$). The BayesR and BayesRC models were implemented using BayesRv2 [83] and BayesRCO [84] software, respectively. Both software programs employed the same parameters, with a total of 25,000 MCMC iterations, of which the first 5,000 iterations were discarded as burn-in.

## Validation of RNA editing

We cultured bovine mammary epithelial cells (MAC-T) in 90% Dulbecco's Modified Eagle's Medium (DMEM) supplemented with 10% heat-inactivated fetal bovine serum (FBS; Vazyme Biotech Co., Ltd., Nanjing, China) at an atmosphere of 5% CO2 and a temperature of 37°C. For the MAC-T cells, we extracted total RNA and DNA using commercial kits (RC112, Vazyme Biotech Co., Ltd., Nanjing, China; and QIAamp DNA Mini Kit, QIAGEN GmbH, Germany). The mRNA was then reverse-transcribed to cDNA using the R333 kit. We randomly selected eight sequences, including those with RNA editing, to design primers (S10 Table) through Primer 3. Polymerase chain reaction (PCR) amplifications were conducted using cDNA and DNA as templates. To identify RNA editing, we sequenced the PCR amplification products using an ABI 3730XL DNA Analyzer (Applied Biosystems, Foster, CA, USA).

## Supporting information

**S1 Fig. The proportion of variance explained by chromosomes and genomic regions in healthy and reproduction traits.** The proportion of variance explained by each chromosome against chromosome length is shown for (A) age at first calving (AFC), (B) cow conception rate (CCR), (C) daughter calving ease (DCE), (D) somatic cell score (SCS), (E) daughter still birth (DSB), and (F) daughter pregnancy rate (DPR) by joint analysis. The numbers in the circles and squares are the chromosome numbers. The regression adjusted R2 (P-value) were -0.029 (0.65) for AFC, 0.044 (0.14) for CCR, -0.037 (0.97) for DCE, 0.074 (0.083) for SCS, 0.013 (0.25) for DSB, and -0.037 (0.95) for DPR, respectively. The variance explanation of each chromosome for heifer conception rate (HCR) was not shown here, due to its Log-likelihood analysis was not converged.
(TIF)

**S2 Fig. The proportion of variance explained by of each class of genomic region for five milk production traits. milk yield (MY), milk fat yield (MFY), milk protein yield (MPY), milk fat percentage (MFP) and milk protein percentage (MPP).**
(TIF)

**S3 Fig. The Pearson correlation between tissues based on their t-statistics.**
(TIF)

**S4 Fig. The proportion of variance explained by different fold-change groups for milk production traits.**
(TIF)

**S5 Fig. The proportion of variance explained by different fold-change groups for milk production traits after correcting *DGAT1* regions.**
(TIF)

**S6 Fig. The proportion of variance explained by the miRNA targets.** The red and blue colors of point represent the down and up-regulated miRNAs. The size of point represents the absolute value of fold change.
(TIF)

**S7 Fig. The Pearson correlation between ADAR/ADARB1 and casein genes based on their expression value.**
(TIF)

**S8 Fig. The chromatogram of DNA and cDNA in *ACACA* (Chr19:13565203–13565656) by Sanger sequencing.** The validated editing sites were marked with yellow background. The novel editing sites were marked with blue background.
(TIF)

**S9 Fig. The chromatogram of DNA and cDNA in *SLC24A5* (Chr10:62245280:62247248) by Sanger sequencing.** The validated editing sites were marked with yellow background. The novel editing sites were marked with blue background.
(TIF)

**S10 Fig. The chromatogram of DNA and cDNA in *ACSS2* (Chr13:64198090–64198291) by Sanger sequencing.** The validated editing sites were marked with yellow background.
(TIF)

**S11 Fig. The chromatogram of DNA and cDNA in downstream of *MDM4* (Chr16:2300794–2301061) by Sanger sequencing.** The validated editing sites were marked with yellow background.
(TIF)

**S12 Fig. The chromatogram of DNA and cDNA in downstream of *GTF3C4* (Chr11:102756753–102757017) by Sanger sequencing.** The validated editing sites were marked with yellow background. The novel editing sites were marked with blue background.
(TIF)

**S13 Fig. The chromatogram of DNA and cDNA in downstream of *SLC7A1* (Chr12:30993903–30994907) by Sanger sequencing.** The validated editing sites were marked with yellow background.
(TIF)

**S14 Fig. The chromatogram of DNA and cDNA in *MAPKAPK5* (Chr17:62255165–62256066) by Sanger sequencing.** The validated editing sites were marked with yellow background.
(TIF)

**S15 Fig. The number of RNA editing sites detected by different numbers of samples.**
(TIF)

**S16 Fig. The proportion of variance explained by differential DNA methylation regions (DMRs) between lactation and non-lactaing period in mammary, blood and brain.**
(TIF)

**S17 Fig. The proportion of variance explained using 671K and 625K genotype panel for five milk production traits.**
(TIF)

**S1 Table. Number, mean and standard deviation (SD) of phenoptype for 12 traits.**
(XLSX)

**S2 Table. The information of 103 biopsy mammary samples used for differentially expressed analysis.**
(XLSX)

**S3 Table. The KEGG results of significantly up-regulated and down-regulated genes.**
(XLSX)

**S4 Table. The proportion of variance explained by the up-regulated and down-regulated lncRNAs for milk production traits.**
(XLSX)

**S5 Table. The enrichment level of the negative correlated mRNAs within predicted targets of the intended miRNA using Fisher's exact test.**
(XLSX)

**S6 Table. The overlapped results between significant enrichment of targets with correlation and the relatively large proportion of variance explained by their targets.**
(XLSX)

**S7 Table. Validated results in eight RNA editing regions using MAC-T cells.**
(XLSX)

**S8 Table. The functional annotation of specific, differential, and common RNA editing sites between two mammary stages.**
(XLSX)

**S9 Table. The information of 6,642 samples used for *t*-statistics computation.**
(XLSX)

**S10 Table. The primers of RNA editing validation.**
(XLSX)

## Acknowledgments

We thank Dr. Hans Daetwyler and Dr. Iona Macleod for their valuable expertise and assistance in statistical analyses. We also thank Dr. Lijun Shi for her valuable expertise and assistance in experiment validation of RNA editing.

## Author contributions

**Conceptualization:** Wentao Cai, Shengli Zhang, Jiuzhou Song.

**Data curation:** Wentao Cai.

**Formal analysis:** Wentao Cai.

**Funding acquisition:** Wentao Cai, Junya Li, Jiuzhou Song.

**Investigation:** Wentao Cai, Jiuzhou Song.

**Methodology:** Wentao Cai, Michael E. Goddard, Jiuzhou Song.

**Project administration:** John B. Cole.

**Resources:** John B. Cole, Junya Li, Jiuzhou Song.

**Supervision:** Shengli Zhang, Jiuzhou Song.

**Validation:** Wentao Cai.

**Visualization:** Wentao Cai.

**Writing – original draft:** Wentao Cai.

**Writing – review & editing:** Jiuzhou Song.

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
