## [Decision Letter · Decision Letter 0]

14 Oct 2024

Dear Dr Song,

Thank you very much for submitting your Research Article entitled 'Mammary Gland Multi-Omics Data Reveals New Genetic Insights into Milk Production Traits in Dairy Cattle' to PLOS Genetics.

The manuscript was fully evaluated at the editorial level and by independent peer reviewers. The reviewers appreciated the attention to an important problem, but raised some substantial concerns about the current manuscript. Based on the reviews, we will not be able to accept this version of the manuscript, but we would be willing to review a much-revised version. We cannot, of course, promise publication at that time.

If you decide to revise the manuscript for further consideration at PLOS Genetics, please aim to resubmit within the next 60 days, unless it will take extra time to address the concerns of the reviewers, in which case we would appreciate an expected resubmission date by email to plosgenetics@plos.org.

If present, accompanying reviewer attachments are included with this email; please notify the journal office if any appear to be missing. They will also be available for download from the link below. You can use this link to log into the system when you are ready to submit a revised version, having first consulted our Submission Checklist .

PLOS has incorporated Similarity Check , powered by iThenticate, into its journal-wide submission system in order to screen submitted content for originality before publication. Each PLOS journal undertakes screening on a proportion of submitted articles. You will be contacted if needed following the screening process.

To resubmit, log into your Editorial Manager account and select the option 'Revise Submission' in the 'Submissions Needing Revision' folder.

We are sorry that we cannot be more positive about your manuscript at this stage. Please do not hesitate to contact us if you have any concerns or questions.

Yours sincerely,

Zoltán Kutalik, PhD

Section Editor

PLOS Genetics

Overall, the reviewers are generally positive about the study, but the submission in this current form does not comply with the publishing policies of PLoS Genetics: Both genotypic and phenotypic data and the program code to analyse these data must be made publicly available.

Reviewer's Responses to Questions

**Comments to the Authors:**

Reviewer #1: The authors present a throughout study of integrating genomics aiming to improve the GS. The collection of molecular phenotype is unique and partially been experimental validate. Even though the improvement of GS is not substantial, the knowledge and information generated during the study is also useful to the community.

Minor comments:

1. Line 30~32, the summary about selection of miRNA is not informative. It is a nice result. However, the wording did not function as abstract, since the reader have to read the result part to understand the sentence. Please rephrase it.

2. Line 36, more than what?

3. Line 37~38, If it is routine but not the SNP panel you just mentioned, "these" is not necessary.

4. Line 78~79, Not, this is not the argument. "Replacing the routine SNP panel with the WGS variants marginally improve the prediction accuracy."

5. Line 79~81, Incomplete sentence.

6. Line 92, have not yet.

7. Line 103, facilitate.

8. Line 116, hypothesized.

9. Line 146~147, go to discussion.

10. Line 148~149, go to discussion.

11. Line 154, estimations were

12. Line 157, More than which type of variants?

13. Line 220, remove the part of sentence with DESeq2, it belong to mthod.

14. Line 540, what difference and which class.

Reviewer #2: Attached review

Reviewer #3: This is a very interesting and well written manuscript. I agree with the author's claim in the title that new insights into the genetics of milk production are generated. However, I do not find the improvement of reliabilities very impressive. I only have one concern related to the description of the methodlogy, especially the data sources. The design is complex using various data. The reader might get lost. I think a graphical overview on the workflow and data would be appreciated. This will also improve the understanding of the results. When e.g. it is stated that there is a efect of particular miRNAs on protein content it would mean that there is a relationship between these miRNA and protein content PTAs, right?

Furthermore, the introduction, although very nicely written, could be shortened a bit.

**Have all data underlying the figures and results presented in the manuscript been provided?**

Reviewer #1: Yes

Reviewer #2: Yes

Reviewer #3: Yes

PLOS authors have the option to publish the peer review history of their article (what does this mean? ). If published, this will include your full peer review and any attached files.

**Do you want your identity to be public for this peer review?** For information about this choice, including consent withdrawal, please see our Privacy Policy .

Reviewer #1: **Yes: ** Zexi Cai

Reviewer #2: No

Reviewer #3: No

---

## [Editor Report · Decision Letter 1]

17 Dec 2024

PGENETICS-D-24-00953R1

Mammary Gland Multi-Omics Data Reveals New Genetic Insights into Milk Production Traits in Dairy Cattle

PLOS Genetics

Dear Dr. Song,

Thank you for submitting your manuscript to PLOS Genetics. After careful consideration, we feel that it has merit but does not fully meet PLOS Genetics's publication criteria as it currently stands. Therefore, we invite you to submit a revised version of the manuscript that addresses the points raised by the Editors.

Please submit your revised manuscript within 30 days Feb 15 2025 11:59PM. If you will need more time than this to complete your revisions, please reply to this message or contact the journal office at plosgenetics@plos.org. Please include the following items when submitting your revised manuscript:

We look forward to receiving your revised manuscript.

Kind regards,

Zoltán Kutalik, PhD

Section Editor

PLOS Genetics

Aimée Dudley

Editor-in-Chief

PLOS Genetics

Anne Goriely

Editor-in-Chief

PLOS Genetics

**Editor Comments** :

Before we can send back the manuscript to the original reviewers, we would like the authors to address our concerns about data and code sharing:

1. In the data availability section, explicitly state where the genotype data can be downloaded from, also where the phenotype (milk yield) can be obtained from.

2. While there is a code provided for the permutation procedure (https://github.com/WentaoCai/Permutation) described in Eq (6), the whole analysis flow is not described in code pipeline. Please provide code to reproduce the whole pipeline, not only a small part of it. Of course, the code can contain system commands to launch GCTA with explicit flags, methylation data processing, RNA editing, DESeq2 pipeline, RNAseq data processing, genotype data processing (PLINK commands), LDAK commands, etc.

**Reviewers' comments:**

**Figure resubmission:**
---

## [Decision Letter · Decision Letter 2]

13 Feb 2025

PGENETICS-D-24-00953R2

Mammary Gland Multi-Omics Data Reveals New Genetic Insights into Milk Production Traits in Dairy Cattle

PLOS Genetics

Dear Dr. Song,

Thank you for submitting your manuscript to PLOS Genetics. After careful consideration, we feel that it has merit but does not fully meet PLOS Genetics's publication criteria as it currently stands. Therefore, we invite you to submit a revised version of the manuscript that addresses the points raised during the review process.

Please submit your revised manuscript within 30 days Mar 15 2025 11:59PM. If you will need more time than this to complete your revisions, please reply to this message or contact the journal office at plosgenetics@plos.org. Please include the following items when submitting your revised manuscript:

We look forward to receiving your revised manuscript.

Kind regards,

Bertrand Servin

Academic Editor

PLOS Genetics

Zoltán Kutalik

Section Editor

PLOS Genetics

Aimée Dudley

Editor-in-Chief

PLOS Genetics

Anne Goriely

Editor-in-Chief

PLOS Genetics

**Reviewers' comments:**

Reviewer's Responses to Questions

**Comments to the Authors:**

Reviewer #1: The authors has addressed all the concern from me.

I have one more additional comment about the abstract.

After the revision, the abstract still difficult to follow. The sentence is swiching among three marjor categories of their results: functional intepretation, method/strategy develped and the contribution to the GS. I would like to ask the authors to reorganize the sentence to describe one at a time and the flow could follow above mentioned order.

Reviewer #2: I have reviewed the manuscript titled ‘Mammary gland multi-omics data reveals new genetic insights into milk production traits in dairy cattle’ by Cai et al. I would like to thank the authors for addressing all my comments.

The model definition for the variance component analysis needs to be improved; it is hard to follow the different models the authors performed in the text and tables (Tables 1 and 2).

Minor comments

L74: Please remove the double comma.

L93-94: Please add blank spaces before the reference (“BayesRC [26], MultiBLUP [27]”).

L435: Please rephrase, starting the sentence with “Histone modifications such as …”

L520: Please add a comma after “i.e.,” and it should be in italics (idem for throughout the text).

L604: Please use the same notation: error variance should be σ_ε^2

L601-613: It is unclear why the authors used model 3 and the difference with model 4. I understand that the authors computed different GRMs and, using model 4, calculated the variance component for each functional class. Was the model 3 was used as a reference? Please clarify in the text.

L605-606: Please rephrase as follows: “The heritability (h^2=(σ_g^2)/(σ_p^2 )) was the proportion of phenotypic variance (σ_p^2) explained by all variants together.”

L614-616: This explanation should be before model 3 and clarify the value of using this model (h_all^2 comes from model 3?).

L621: Please check if the abbreviation should be in bold. Usually, only vectors and matrices are in bold.

L661: Please rephrase as follows: “We performed BayesR and BayesRC using the following model:”

L664: Please add a blank space after the comma.

L677-687: This paragraph should be removed because it is not considered anymore in the results and discussion.

Table 2: Please check the total values. Some of them are not the exact sum of each row.

**Have all data underlying the figures and results presented in the manuscript been provided?**

Reviewer #1: **No: ** The genotype and phenotype data is still not public avaliable.

Reviewer #2: Yes

PLOS authors have the option to publish the peer review history of their article (what does this mean? ). If published, this will include your full peer review and any attached files.

**Do you want your identity to be public for this peer review?** For information about this choice, including consent withdrawal, please see our Privacy Policy .

Reviewer #1: **Yes: ** Zexi Cai

Reviewer #2: No

**Figure resubmission:**
---

## [Editor Report · Decision Letter 3]

10 Mar 2025

PGENETICS-D-24-00953R3

Mammary Gland Multi-Omics Data Reveals New Genetic Insights into Milk Production Traits in Dairy Cattle

PLOS Genetics

Dear Dr. Song,

We will be ready to accept your manuscript for publication in PLoS Genetics, provided you answer the comments below by the academic editor.

Thank you for submitting your manuscript to PLOS Genetics. After careful consideration, we feel that it has merit but does not fully meet PLOS Genetics's publication criteria as it currently stands. Therefore, we invite you to submit a revised version of the manuscript that addresses the points raised during the review process.

Please submit your revised manuscript within 30 days Apr 09 2025 11:59PM. If you will need more time than this to complete your revisions, please reply to this message or contact the journal office at plosgenetics@plos.org. Please include the following items when submitting your revised manuscript:

We look forward to receiving your revised manuscript.

Kind regards,

Bertrand Servin

Academic Editor

PLOS Genetics

Zoltán Kutalik

Section Editor

PLOS Genetics

Aimée Dudley

Editor-in-Chief

PLOS Genetics

Anne Goriely

Editor-in-Chief

PLOS Genetics

**Additional Editor Comments:**

## Main comments

- The Abstract still requires careful editing, in particular:

line 20-21: *defined functional classes* , what do you mean by "define", functional classes of what objects / elements ? please be more specific

line 26: *Mammary specific genes* : specify what is meant by this

line 28: *Differential expression* between what ?

line 30: *exhibiting enriched genetic variance* not clear what this means

line 32 *DNA methylation changes <0.2 contribute the most variance* please rephrase

lines 34: ~~compared to randomly shifted sites~~ replace by *than expected by chance*

line 35: *candidate miRNAs* candidate for what ?

lines 35-37: *identifying overlaps ... explained by these targets* in the context of the abstract this sentence cannot be understood, please simplify

line 37: *a functional annotation framework* what do you mean ?

- A recurring problem throughout the manuscript is the confusion betwen "genetic variance", "variance" and "proportion of variance explained". These terms are not exchangeable (*e.g.* 0.2 genetic variance is not the same as a 20% proportion of genetic variance). Phenotypic variance and genetic variance are not equal etc. Please be specific in the text, figure legends, axis labels etc. as to which is the correct term.

- In figures, when plotting the enrichment of categories, please present the result on a log-scale as an "enrichment" of 0.5 is of the same magnitude as an enrichment of 2 (see for example figure 2F, 3E, 4B etc.)

- Please upload the computer codes in an archived form and provide the identifier from a repository such as SoftwareHeritage, Dryad or Zenodo.

- The Methods section *Mammary-specific gene expression* is not clear enough. Please recall what are the tissue samples categories and the exact null hypotheses tested.

## Specific comments

line 88-89 : "highest proportion of the variance per SNP" this is not clear, please explain

line 91: *variant in functional class* rephrase

line 96 : delete not

line 98 99: rephrase

line 109 : *be irrelevant* come from irrelevant ?

lines 111: *tissue* delete, a tissue is not an organ

line 140: *variance explained by MFP* -> variance **of** MFP (see main comment above)

line 145: *intergenic and intron* -> intergenic regions and introns / intergenic and intronic regions

line 154: t-statistic -> the type of statistic matters less than the null hypothesis tested. Please explain what hypotheses are tested here (see above).

line 159 : *We observed highly positive correlations* correlation of what ? (also line 161)

line 161-162: *We integrated tissue specific ...* this sentence is not clear, be more specific

line 171-175: please specificy the FDR threshold and the method used to estimate it.

line 185: *explain large variance* see main comment

line 190: *the other two* -> *two other*

line 191: *according* -> for line 208-209: *The variances ...* this is not a valid sentence, please explain.

lines 215: *inverse correlation* do you mean negative ?

lines 244-247: *We proposed the candidate ...* These two sentences are not clear, please rephrase.

line 293: *more genetic variance* than what ?

line 384 *more closed* closer ?

line 387: *milk cell is moderately correlated with lactating mammary gland based on their t-statistics* I don't understand this sentence, please explain and rephrase.

line 391: *The liver captured considerable variance* The organ certainly does not. What do you mean ?

line 397: *immune, disease resistant* -> *immunity, disease resistance*

line 403-404: *thousands of lncRNAs are dynamic changes* please rephrase, this does not make sense

line 408: *As the simple target ...* This sentence is not finished

line 525: *MicoRNA* -> MicroRNA

**Figure resubmission:**
---

## [Editor Report · Decision Letter 4]

3 Apr 2025

Dear Dr Song,

We are pleased to inform you that your manuscript entitled "Mammary Gland Multi-Omics Data Reveals New Genetic Insights into Milk Production Traits in Dairy Cattle" has been editorially accepted for publication in PLOS Genetics. Congratulations!

Yours sincerely,

Bertrand Servin

Academic Editor

PLOS Genetics

Zoltán Kutalik

Section Editor

PLOS Genetics

Aimée Dudley

Editor-in-Chief

PLOS Genetics

Anne Goriely

Editor-in-Chief

PLOS Genetics

Comments from the reviewers (if applicable):

**Data Deposition**

http://datadryad.org/submit?journalID=pgenetics&manu=PGENETICS-D-24-00953R4

**Press Queries**

---

## [Editor Report · Acceptance letter]

PGENETICS-D-24-00953R4

Mammary Gland Multi-Omics Data Reveals New Genetic Insights into Milk Production Traits in Dairy Cattle

Dear Dr Song,

We are pleased to inform you that your manuscript entitled "Mammary Gland Multi-Omics Data Reveals New Genetic Insights into Milk Production Traits in Dairy Cattle" has been formally accepted for publication in PLOS Genetics! Your manuscript is now with our production department and you will be notified of the publication date in due course.

With kind regards,

Anita Estes

PLOS Genetics

On behalf of:
